# Delving into Large Language Models for Effective Time-Series Anomaly Detection

**Junwoo Park**[1,2], **Kyudan Jung**[1,2], **Dohyun Lee**[1,2], **Hyuck Lee**[2], **Daehoon Gwak**[1],
**ChaeHun Park**[1], **Jaegul Choo**[1], **Jaewoong Cho**[2]
[1]KAIST AI    [2]KRAFTON
{junwoo.park,kyudan,aiclaudev,daehoon.gwak,ddehun,jchoo}@kaist.ac.kr
{dlgur0921,jwcho}@krafton.com

## Abstract

Recent efforts to apply Large Language Models (LLMs) to time-series anomaly detection (TSAD) have yielded limited success, often performing worse than even simple methods. While prior work has focused solely on downstream performance evaluation, the fundamental question—*why do LLMs struggle with TSAD?*—has remained largely unexplored. In this paper, we present an in-depth analysis that identifies two core challenges in understanding complex temporal dynamics and accurately localizing anomalies. To address these challenges, we propose a simple yet effective method that combines statistical decomposition with index-aware prompting. Our method outperforms 21 existing prompting strategies on the AnomLLM benchmark, achieving up to a 66.6% improvement in F1 score. We further compare LLMs with 16 non-LLM baselines on the TSB-AD benchmark, highlighting scenarios where LLMs offer unique advantages via contextual reasoning. Our findings provide empirical insights into how and when LLMs can be effective for TSAD. The code is publicly available at: https://github.com/junwoopark92/LLM-TSAD.

## 1 Introduction

Large language models (LLMs) have shown impressive zero-shot numerical reasoning capabilities across various tasks, including math problem-solving [28, 41], tabular analysis [19, 27, 55, 66], and time series forecasting [26, 54, 68]. These numerical reasoning tasks typically require interpreting numbers embedded in a natural language prompt, performing arithmetic or logical operations, and accurately understanding numerical relationships within their domain context. *Time series anomaly detection (TSAD)* similarly involves these capabilities, as it requires interpreting numerical deviations and recognizing temporal patterns within time series. Motivated by these similarities, recent studies [4, 20, 45, 79, 83, 87, 89] have shown the potential of LLMs for TSAD, highlighting their strengths in handling diverse input formats and leveraging domain-specific contexts.

However, while demonstrating the potential of LLMs for TSAD, these studies simultaneously reveal significant limitations. Specifically, LLMs frequently exhibit false positive detections when time series are presented in textual form, often misclassifying normal fluctuations or pervasive background noise as anomalies [4, 20]. As an alternative, graphical plots have been used; however, LLMs often only identify prominent spikes or visually salient anomalies, missing more nuanced anomalies [79, 89]. These failure cases have led to a pessimistic view of LLMs' capabilities at TSAD. However, prior work has focused predominantly on evaluating downstream detection accuracy, without addressing the fundamental question: *why do LLMs struggle with TSAD?* The underlying reasons, whether due to input representations, temporal modeling, or prompting strategies, remain unclear.

To systematically investigate these unresolved questions, we begin by decomposing the TSAD problem into two essential subtasks: (1) distinguishing between normal and anomalous patterns (time-series understanding), and (2) identifying the precise anomaly intervals (anomaly localization).

39th Conference on Neural Information Processing Systems (NeurIPS 2025).

Unlike prior works that treat TSAD as a monolithic task [4, 20, 79, 89], this decomposition allows us to isolate and diagnose the distinct failure modes of LLMs.For the first subtask, our preliminary analysis reveals that LLMs often miss subtle anomalies when complex normal patterns, such as overlapping trend and seasonality, obscure their presence. For the second subtask, we observe that widely used prompts implicitly require LLMs to count sequence tokens to pinpoint anomaly positions—a capability that LLMs consistently struggle with [22, 78, 85]. This task decomposition thus surfaces fine-grained limitations of LLMs that remain hidden under aggregate performance metrics.

These observations directly inform our solution, which explicitly targets both limitations. To address the difficulty of detecting anomalies obscured by complex normal patterns, we apply statistical decomposition to remove only the seasonal component (*i.e.,* de-seasonalization). Since seasonal patterns exhibit consistent, repeating structure, they are typically treated as normal rather than anomalous. Removing seasonal patterns enhances the visibility of subtle anomalies that are concealed within complex normal dynamics, thus facilitating accurate detection by LLMs. To overcome LLMs' struggles with position reasoning via token counting, we design index-aware prompting that explicitly embeds positional cues in the input. This approach highlights the often-overlooked utility of time-series text, in contrast to prior methods that predominantly rely on visual inputs.

As a result, on the AnomLLM benchmark [89], our method surpasses all 21 existing prompting strategies, boosting F1 scores by an average of 39.9% and up to 66.6% across both open-source and proprietary LLMs. Ablation studies further validate the critical role of both statistical decomposition and explicit indexing. Finally, building on these improvements, we revisit a question often left unaddressed in earlier work: LLMs' practical values in TSAD compared to conventional numerical methods. We benchmark our method against 16 non-LLM methods from the TSB-AD suite [46], comparing zero-shot accuracy and inference efficiency. We also explore scenarios where the contextual reasoning capabilities of LLMs offer practical benefits, such as selectively excluding known or explainable anomalies by domain-specific contexts.

Our main contributions are as follows:

- We present a comprehensive analysis of the fundamental limitations of LLMs in TSAD, explicitly identifying the challenges in understanding temporal patterns and accurately localizing anomalies.

- To address these limitations, we propose a simple yet effective method that combines statistical decomposition with index-aware prompting, significantly improving the average F1 score by up to 66.6% compared to existing LLM-based TSAD methods.

- Based on the improved performance, we evaluate the practical justification for employing LLMs in TSAD by comparing them with diverse non-LLM methods, highlighting specific scenarios in which the contextual reasoning capabilities of LLMs offer tangible advantages.

## 2 Analysis: why do LLMs struggle with TSAD?

Throughout our study, we build on the experimental setup proposed in AnomLLM [89], which systematically categorizes recent prompt designs and anomaly types. For clarity and due to space constraints, we provide detailed descriptions of representative TSAD methods and other LLM-based methods in the related work section in Appendix A.1.

**Task Setup.** In the LLM-based TSAD, an input time series $\mathbf{x}_{t-L:t} = (x_{t-L}, \ldots, x_t)$ is first converted into a comma-separated list of normalized values (time-series text) or a simple line plot (time-series image). These representations are then fed into an LLM, guided by natural language instructions such as: "Identify all contiguous intervals $[a_i, b_i]$ that exhibit anomalous behavior." The LLM generates a free-form textual response, which is post-processed to extract interval predictions of the form $\{(a_1, b_1), (a_2, b_2), \ldots, (a_{N_t}, b_{N_t})\}$, where each $(a_i, b_i)$ denotes an anomaly span. This formulation differs from conventional TSAD methods, which compute a scalar anomaly score for each timestamp followed by thresholding.

**Analysis of common failure cases.** We begin with an analysis of AnomLLM failure cases. Figure 1 shows two cases that both exhibit an F1 score of zero but reflect distinct types of failure in the anomaly detection process: (a) the LLM fails to detect any anomaly, despite the presence of a trend shift anomaly; (b) the LLM produces a prediction; however, the predicted anomaly interval does

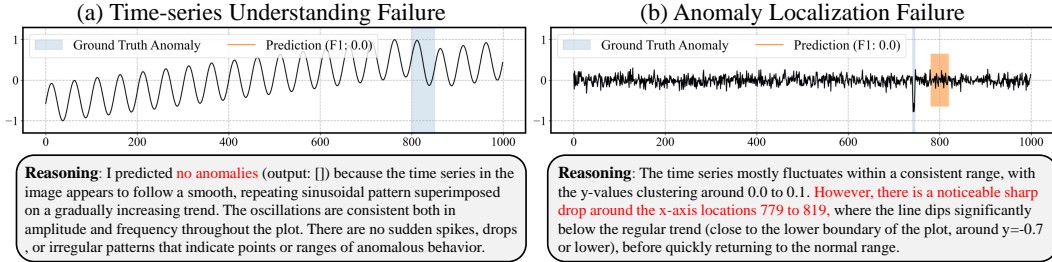

Figure 1: Two representative failure cases of LLM-based TSAD. Although both cases result in an zero F1 score, the reasoning of LLM outputs reveal that the causes stem from different limitations: one in understanding the time series and the other in localization ability.

not overlap with the ground truth, resulting in a zero score. Interestingly, when asked to explain its reasoning in case (b), the model correctly attributes the anomaly to a sudden increase, indicating a successful understanding of the time series pattern. This suggests that the failure in case (b) lies primarily in the localization step, not in understanding. On the other hand, in case (a), the model concludes that there are no anomalies at all, implying a fundamental failure in recognizing the anomaly in the first place.

These examples highlight two key challenges in LLM-based zero-shot anomaly detection: (1) understanding time-series dynamics, and (2) accurately localizing the anomaly intervals. To isolate these two types of failure, we conducted two experiments: instance-level TSAD (Section 2.1) which focuses on the understanding failures (a) and TSAD with Ground Truth (GT) anomalies (Section 2.2) which targets the localization failures (b).

## 2.1 Evaluating LLMs' time-series understanding capability

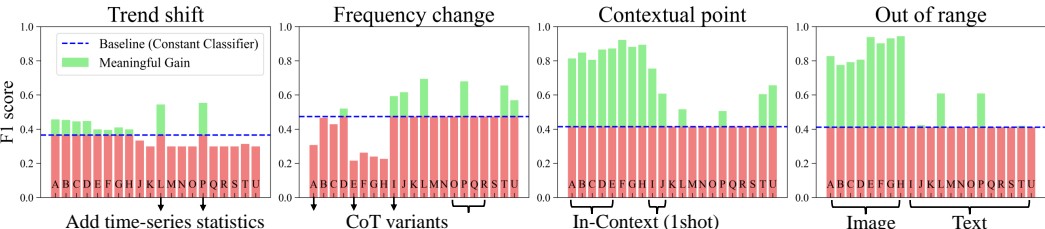

Figure 2: Instance-level TSAD results of 21 prompts (A-U), including two sequence representation (image: A-H and text: I-U), CoT (A,E,I,O-R), and In-Context (A-D, I-J) strategies (details in Table 6) on four anomaly types to estimate understanding capability of LLMs. Even with these prompts, LLMs still struggle to detect anomalies such as trend shifts and frequency changes.

In TSAD, final performance is ultimately calculated by either sequence-length binary labels or interval predictions, inevitably depending on localization capability. Thus, we relax the original problem into a simpler formulation, deciding only whether any anomaly exists in the entire sequence without precise localization, which we refer to as instance-level anomaly detection. This binary detection task is inherently easier than identifying multiple anomalous intervals in the sequence. Therefore, if a model fails to even detect the presence of an anomaly, it clearly lacks the time-series understanding necessary for TSAD. Experimental details including the task prompt can be found in Appendix D.1.

Figure 2 illustrates the varying degrees of time-series understanding exhibited by LLMs across different anomaly types. For trend and frequency anomalies—which involve subtle and intertwined changes that closely resemble normal patterns—LLMs perform only marginally better, or even worse, than a trivial baseline that always predicts the same label (normal or abnormal) regardless of the input. In contrast, for point and range anomalies, LLMs using sequence image prompts (A–H) achieve near-perfect accuracy, unlike those relying on the time-series text. Overall, despite employing prompt strategies such as Chain of Thought (CoT) and various in-context exemplars—effective in many NLP tasks—LLMs exhibit limited capability to understand temporal dynamics.

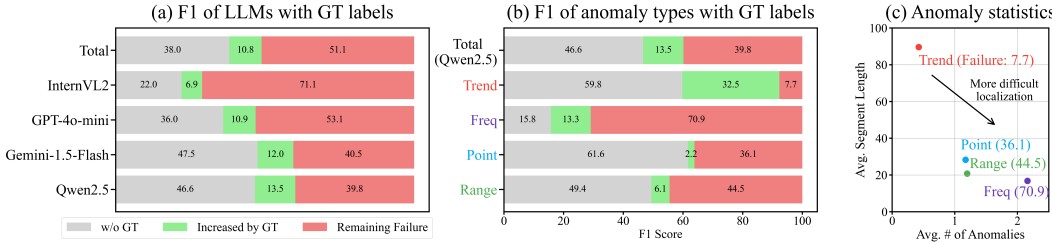

Figure 3: TSAD results with GT anomaly labels to estimate localization failure. (a) shows the F1 improvement by GT labels for each model, (b) shows the improvement by GT labels for each anomaly type, and (c) illustrates the relationship between the number and length of anomalies per type and the remaining failure.

## 2.2 Evaluating LLMs' time-series localization capability

To determine whether the limited performance of LLMs originates from poor localization, we design a localization-only experiment that explicitly removes the need for time-series understanding. Specifically, we highlight GT anomaly regions within the time-series image (e.g., (a) and (b) in Figure 1). Apart from this visual cue, the task setup remains identical to the original TSAD setting: the model must return anomaly intervals along the x-axis. For clarity, we use time-series images here, but Section 4.2 also evaluates localization on time-series text. The prompt and additional details for this experiment are provided in the Appendix D.2.

In this configuration, the location of anomalies is explicitly given within the input image, and the model is only required to translate this information into precise coordinate predictions. If localization were not a bottleneck, we would expect that LLMs could achieve near-perfect scores.

However, as shown in Figure 3(a), although all models benefit from this GT-provided setting, the performance gains remain limited. This suggests that the bottleneck lies not only in identifying where anomalies occur but also in precisely localizing those regions. In other words, even when LLMs are given highlighted regions, they often struggle to translate them into accurate output, indicating a core limitation in localization capability.

Beyond the overall performance plateau, a closer look at individual anomaly types reveals further insights into localization difficulty. As shown in Figure 3(b), trend and frequency anomalies—previously the most challenging for understanding (Figure 2)—exhibit larger F1 improvements than point and range anomalies. This disparity suggests that localization difficulty is not uniform but is closely tied to the characteristics of each anomaly type. Figure 3(c) reinforces this interpretation: residual error rates correlate with the number and fragmentation of anomaly segments. Frequency anomalies, in particular, often consist of many scattered segments, making precise localization substantially harder.

## 3 Methodology

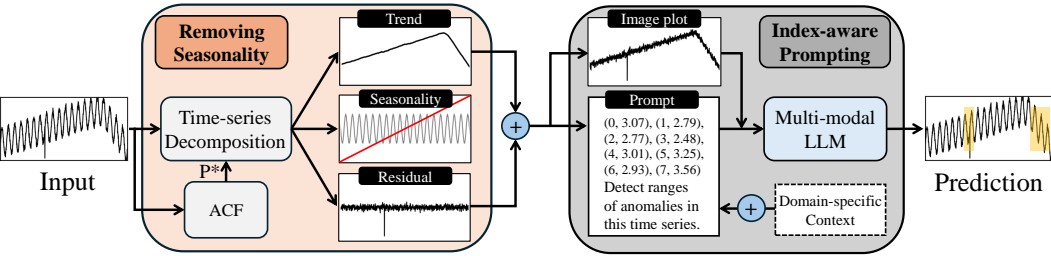

Figure 4: An overview of our proposed method to mitigate understanding and localization challenges.

From the two preceding experiments, we confirmed that the current performance limitations of LLMs in TSAD stem from both the understanding and localization challenges. Understanding complex temporal structures has long been a central challenge in time-series analysis. A representative strategy is to decompose time series into structural components such as trend, seasonality, and residuals [5]. Existing non-LLM methods [53, 74, 75] have often embedded this decomposition internally within

models through end-to-end learning pipelines. However, for LLMs with strong pretrained reasoning abilities, training them to perform decomposition may degrade their pretrained capabilities and introduce inefficiencies, particularly due to the need to decode all outputs as text compared to the use of numerical algorithms.

Motivated by recent advances in tool-augmented LLM approaches [30, 56], we introduce a simple yet effective method that enhances LLM-based TSAD by leveraging lightweight external tools for time-series decomposition. This design allows the LLM to focus on its core strength—reasoning over structured prompts—while benefiting from accurate, preprocessed inputs, as illustrated in Figure 4.

## 3.1 De-seasonalization: supporting LLMs' understanding of time series

Pronounced seasonality often obscures subtle *trend shifts* and *frequency changes*, causing LLMs to overlook anomalies that become obvious once the seasonal cycle is removed. We therefore insert a lightweight de-seasonalization module directly before the LLM-based detector:

**Step 1:** **Period discovery**: Given an input time series $x_{1:T}$, compute the sample autocorrelation [37] function (ACF) $\hat{\rho}$ for lags $k = 1, \ldots, \lfloor T/2 \rfloor$ and identify the dominant period

$$P^\star = \arg \max_{k \geq 1} \hat{\rho}(k).$$

**Step 2:** **Additive decomposition**: Using classic decomposition [5, 12] with window length $P^\star$, decompose $x_t = s_t + \tau_t + r_t$, where $s_t$ is seasonality, $\tau_t$ the trend, and $r_t$ the residual.

**Step 3:** **Residual pass-through**: Feed the de-seasonalized sequence $\tilde{x}_t = x_t - s_t$ to the LLM as both a time-series text and image.

In this process, global autocorrelation is used to estimate the dominant period, enabling the decomposition to remove only stable, recurring seasonality, while preserving subtle deviations like trend or frequency changes for LLM-based detection.

From a cost perspective, de-seasonalization can be efficiently implemented using a fast Fourier transform, reducing the complexity to $O(T \log T)$, and incurring negligible overhead relative to LLM inference. More sophisticated methods, such as STL [58], Seasonal-HODMD [40], and Prophet [69] can also be integrated with modest additional overhead and without altering the pipeline. Even this lightweight preprocessing step alone yields consistent improvements in F1 score across all evaluated LLMs, as demonstrated in Table 3.

Although de-seasonalization using an external algorithm clearly enhances performance, it raises two key questions. First, might LLMs already possess an implicit ability to perform such a decomposition without the need for explicit tools or additional training? Second, in a truly tool-augmented framework, should not an LLM be able to determine whether such preprocessing is necessary? These questions invite a closer examination of the boundary between what LLMs can do inherently and where external tools remain essential. We explore this boundary in depth in Section 4.2.

## 3.2 Index-aware prompting: challenges in positional reasoning over time-series text

The localization problem differs depending on whether the time series is presented as an image or text. When using a time-series image alone, limited index visibility along the x-axis constrains LLMs to infer anomaly intervals from relative visual positions. On the other hand, when using time-series text, existing AnomLLM prompts typically present only a list of numerical values without any explicit positional information. As a result, the LLM must implicitly perform a counting task to infer the positions of the anomalies. However, counting tokens or semantic chunks is a well-documented challenge for

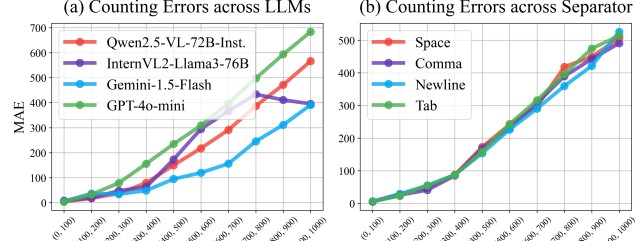

Figure 5: Counting performance of LLMs, quantified by Mean Absolute Error (MAE), across various sequence length intervals. (a) Comparison across LLMs. (b) Comparison across four separator tokens.

LLMs [85]. To assess this limitation for the time-series domain, we designed a simple task that requires counting the number of values in a numerical sequence. As shown in Figure 5, both open-source and proprietary LLMs perform surprisingly poorly on this simple task, even when various separator tokens are used.

To alleviate the counting burden, we introduce index-aware prompting, in which each value in the input sequence is explicitly paired with its positional index. This design allows the LLM to directly return the start and end indices of an anomalous segment once detected, eliminating the need for internal counting and effectively transforming a difficult counting task into a simple lookup. Moreover, since visualizing indices in time-series plots often leads to clutter and overlap, expressing positional information in textual form becomes more practical. This highlights the previously underappreciated utility of time-series text in TSAD.

While we emphasize the use of index-aware prompting to improve anomaly localization, incorporation of index information is not entirely novel. Liu et al. [45] incorporated time-series text with indices. However, most prior works [20, 79, 89] have not utilized index information, possibly due to concerns that longer and more complex prompts could negatively impact task accuracy. To better understand this trade-off, we revisit the issue through empirical evaluation in Section 4.2.

## 4 Experiment

**Datasets.** We leverage the well-curated datasets introduced by AnomLLM [89], which include four representative anomaly types—*point*, *range*, *trend*, and *frequency*—with accurately annotated intervals. These datasets are designed to facilitate fine-grained evaluation under various anomaly conditions. To complement these controlled settings, we also incorporate the TSB-AD-U benchmark [46], which reflects unsupervised anomaly detection in more realistic time-series applications. Certain prompting methods [45, 89] incorporate domain-specific context that characterizes the type of anomaly being tested, whereas our method does not utilize any such context in these benchmarks. The dataset configurations follow the original benchmark and are summarized in Appendix B.2.

**Metric.** AnomLLM [89] uses affiliation metrics [31] as its main evaluation metric, as it offers the advantage of distance-based correction. However, this metric is less sensitive to localization errors and does not clearly differentiate between methods. To ensure a more rigorous evaluation, we therefore employ both the standard and the affiliation metrics.

**Baselines.** We perform experiments using four representative multi-modal LLMs, two of which are open-sourced: Qwen2.5-VL-72B-Instruct and InternVL2-Llama3-76B and two of which are proprietary: GPT-4o (and mini) and Gemini-1.5-Flash. We replicate the 21 prompt variants from the AnomLLM benchmark [89], covering text vs. vision modalities, input formats, and reasoning styles. For comparison with conventional TSAD models in Section 5, we also report results from 16 non-LLM baselines such as ML baselines (SR [60], IForest [44], Sub-PCA [2], KMeans-AD [82], MatrixProfile [84]), DL baselines (CNN [49], USAD [7], AnomalyTransformer [77], OmniAnomaly [65], AutoEncoder [62], TimesNet [75], FITS [80]) Foundation Model (FM) baselines (Chronos [6], TimesFM [18], Lag-Llama [57], MOMENT [24]) with three thresholding methods: Percentile [13, 42], MAD [29], EVT-POT [63, 67] to convert anomaly scores to binary labels.

### 4.1 Comparison with various prompting strategies in LLM-based TSAD

**4.1.1 Main results** Table 1 shows that our proposed method substantially improves to TSAD performance across a wide range of LLMs—InternVL-2, Qwen-2.5, Gemini-1.5, and GPT-4o. In every case, both the standard and affiliation-based precision, recall, and F1 scores rise well above those of existing prompt baselines. Earlier methods often struggled to surpass naive baselines and occasionally underperformed them. In contrast, our method delivers clear, consistent gains that are not merely relative improvements among weak baselines, but absolute advances that surpass the naive baseline by a significant margin.

When comparing our *0shot-text* prompt with that of AnomLLM, we observe a substantial performance gain, even under identical input conditions. While combining vision and text can offer advantages, our results demonstrate that text alone, when combined with techniques such as de-seasonalization and index-aware prompting, can effectively support TSAD and challenge the notion that time series text is inherently limited for TSAD.

Table 1: TSAD results on the AnomLLM benchmark. We report standard and affiliation metrics. "Avg. Diff." denotes the average difference from the baseline, helping to reveal meaningful gains beyond dataset bias. Cells are shaded green/red to indicate values above/below the baseline. The comparison with the other 18 prompts is provided in Appendix E.

| LLMs | Prompt Methods | Standard Metrics | | | | Affiliation Metrics | | | |
|---|---|---|---|---|---|---|---|---|---|
| | | Prec. | Recall | F1 | Avg. Diff. | Prec. | Recall | F1 | Avg. Diff. |
| Naive baseline (Always Normal Predictor) | | 31.75 | 31.75 | 31.75 | – | 31.75 | 31.75 | 31.75 | – |
| InternVL2 (LLaMA3-76B) | AnomLLM (0shot-Text) | 12.94 | 22.15 | 13.38 | -15.59 | 22.12 | 28.38 | 24.10 | -6.88 |
| | AnomLLM (0shot-Vision) | 22.33 | 46.19 | 23.92 | -0.94 | 50.93 | 60.94 | 55.53 | 24.05 |
| | AnomLLM (1shot-Vision-CoT) | 33.72 | 36.28 | 33.67 | 2.81 | 53.62 | 55.55 | 53.93 | 22.62 |
| | Our (0shot-Text) | 27.67 | 52.29 | 29.21 | 4.64 | 54.31 | 60.71 | 55.04 | 24.94 |
| | Our (0shot-Text-Vision) | 30.00 | 59.35 | 34.66 | 9.59 | 58.14 | 71.79 | 62.90 | 32.53 |
| Qwen-2.5 (VL-72B-Inst.) | AnomLLM (0shot-Text) | 25.99 | 25.66 | 25.49 | -6.04 | 45.69 | 42.44 | 43.11 | 12.00 |
| | AnomLLM (0shot-Vision) | 45.78 | 51.00 | 46.65 | 16.06 | 69.50 | 68.99 | 68.78 | 37.34 |
| | AnomLLM (1shot-Vision-CoT) | 27.09 | 29.08 | 27.23 | -3.95 | 39.52 | 40.07 | 39.38 | 7.91 |
| | Our (0shot-Text) | 72.47 | 54.09 | 56.13 | 29.15 | 82.91 | 78.40 | 78.96 | 48.34 |
| | Our (0shot-Text-Vision) | 72.95 | 69.78 | 67.56 | 37.31 | 84.51 | 81.52 | 81.87 | 50.88 |
| Gemini-1.5 (Flash) | AnomLLM (0shot-Text) | 2.01 | 2.47 | 1.89 | -29.63 | 24.90 | 25.58 | 24.14 | -6.88 |
| | AnomLLM (0shot-Vision) | 39.60 | 59.62 | 43.56 | 15.84 | 62.30 | 63.00 | 62.32 | 30.79 |
| | AnomLLM (1shot-Vision-CoT) | 35.61 | 37.85 | 35.20 | 4.47 | 66.06 | 61.92 | 62.45 | 31.73 |
| | Our (0shot-Text) | 57.81 | 43.78 | 46.62 | 17.65 | 66.16 | 64.33 | 64.12 | 33.12 |
| | Our (0shot-Text-Vision) | 68.29 | 65.81 | 63.31 | 34.05 | 84.12 | 81.94 | 82.05 | 50.95 |
| GPT-4o | AnomLLM (0shot-Text) | 19.69 | 17.62 | 17.76 | -13.39 | 46.35 | 45.51 | 44.73 | 13.78 |
| | AnomLLM (0shot-Vision) | 39.54 | 50.60 | 42.03 | 12.31 | 62.19 | 61.99 | 61.68 | 30.20 |
| | AnomLLM (1shot-Vision-CoT) | 31.48 | 38.50 | 32.93 | 2.55 | 56.43 | 55.84 | 55.39 | 24.14 |
| | Our (0shot-Text) | 65.52 | 53.45 | 54.40 | 26.04 | 75.65 | 73.88 | 73.47 | 42.58 |
| | Our (0shot-Text-Vision) | 76.94 | 68.42 | 70.04 | 40.05 | 83.10 | 78.85 | 80.11 | 48.94 |

Table 2: Comparison of TSAD performance gains across LLMs and their general reasoning ability benchmarks. The TSAD improvements positively correlate with general reasoning ability, as models with higher MMLU-Pro/MMMU scores (e.g., GPT-4o, Qwen2.5) yield substantial F1 gains, whereas InternVL2-LLaMA3-76B exhibits only marginal improvement.

| LLMs | GPT-4o | Qwen2.5-VL-72B-Inst. | Gemini-1.5-Flash | InternVL2-LLaMA3-76B |
|---|---|---|---|---|
| **TSAD (F1) – AnomLLM** | 42.03 | 46.65 | 43.56 | 33.67 |
| **TSAD (F1) – Ours** | 70.04 | 67.56 | 63.31 | 34.66 |
| Δ TSAD (Ours − AnomLLM) | **+28.01** | +20.91 | +19.75 | +0.99 |
| **MMMU (Visual Reasoning)** | **69.10** | 64.50 | 56.10 | 58.20 |
| **MMLU-Pro (Language Reasoning)** | **74.68** | 71.59 | 64.09 | 56.20 |

**4.1.2 Model-wise performance discrepancy and analysis of underlying factors** While our proposed prompting strategy improved performance across all evaluated LLMs, the magnitude of improvement varied notably among models. GPT-4o, Qwen2.5-VL-72B-Instruct, and Gemini-1.5-Flash showed significant gains, whereas InternVL2-LLaMA3-76B achieved only marginal improvement. To explore this discrepancy, we examined whether general reasoning ability correlates with improvement in TSAD. We compared models' scores on MMLU-Pro [1] (language reasoning) and MMMU [2] (visual reasoning) benchmarks with their TSAD results. As shown in Table 2, the models with stronger reasoning capabilities, such as GPT-4o (MMLU 74.68, MMMU 69.10) and Qwen (71.59, 64.50), exhibited larger F1 improvements (+28.01 and +20.91), while LLaMA3 (MMLU 56.20, MMMU 58.20) improved by only +0.99. This trend suggests that high general reasoning ability contributes to more effective anomaly detection. Given the computational cost of full TSAD evaluations, aggregate reasoning benchmarks may offer a practical, lightweight proxy for estimating TSAD capability.

## 4.2 In-depth analysis

**4.2.1 Ablation study** Table 3 shows that each component of the proposed method plays a meaningful and complementary role in improving LLM-based TSAD. Removing the de-seasonalization step consistently leads to a drop of 5.64 F1 points on average across the three LLMs, confirming that

---

[1] https://huggingface.co/spaces/TIGER-Lab/MMLU-Pro

[2] https://mmmu-benchmark.github.io/

Table 3: Results of the ablation study on our method, evaluated with standard metrics.

| LLMs | GPT-4o | | | Qwen2.5-VL-72B-Inst. | | | Gemini-1.5-Flash | | | Avg. |
| Metric | Prec. | Recall | F1 | Prec. | Recall | F1 | Prec. | Recall | F1 | Drop |
|---|---|---|---|---|---|---|---|---|---|---|
| **Ours (Full)** | **76.94** | **68.42** | **70.04** | **72.95** | **69.78** | **67.56** | **68.29** | 65.81 | **63.31** | - |
| w/o De-seasonalization | 70.76 | 64.34 | 65.78 | 63.99 | 62.89 | 61.46 | 62.86 | 60.86 | 59.44 | -5.64 |
| w/o Index of time-series text | 41.03 | 43.99 | 41.13 | 38.67 | 43.49 | 38.73 | 49.03 | **67.69** | 52.90 | -23.36 |
| w/o Value of time-series text | 41.06 | 46.25 | 44.47 | 72.47 | 54.09 | 56.13 | 45.66 | 61.94 | 49.11 | -16.88 |
| w/o Time-series image | 65.52 | 53.45 | 54.40 | 65.52 | 53.45 | 54.40 | 57.82 | 43.78 | 46.62 | -14.24 |

eliminating dominant seasonal patterns enables the model to better capture subtle trend and frequency shifts. In particular, the exclusion of index information from the time-series text results in the most severe performance degradation, indicating that LLMs struggle with implicit positional reasoning and significantly benefit from explicit indexing for accurate localization. Omitting either the value information or the sequence image leads to a performance drop, indicating that each modality provides complementary cues. Their combination consistently enhances detection accuracy, suggesting that LLMs effectively integrate textual and visual signals for more precise anomaly detection.

**4.2.2 Can LLMs perform time-series decomposition without external tool?**   The above question holds merit in evaluating the time-series understanding capabilities of LLMs and guiding the design of corresponding detection frameworks that decide whether or not to invoke external tool or not. We assess the decomposition ability of LLMs through two tasks: (1) detecting the presence of components and (2) generating each component's sequence individually.

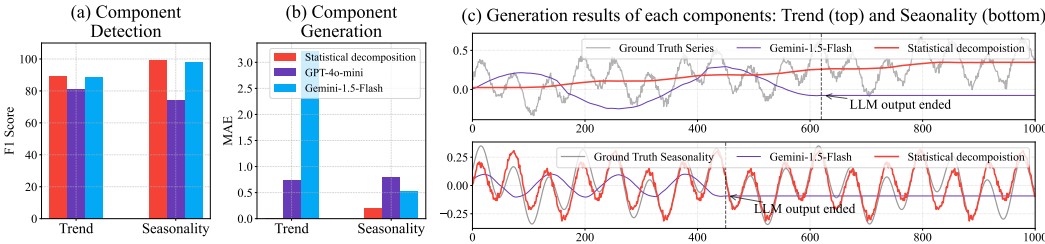

Figure 6: Comparison of LLMs and statistical decomposition for time-series component analysis. (a) shows component detection results, highlighting how well LLMs identify underlying component compared to statistical decomposition. (b) presents quantitative and (c) qualitative results evaluating the accuracy and fidelity of components generated by the LLM. The prompts used in this experiment are illustrated in Figure 17.

Figure 6 shows that LLMs perform comparably to statistical decomposition in identifying the presence of individual components. However, their performance deteriorates significantly when generating sequences for each component individually, especially for long time series, where qualitative examples reveal notable errors and failure cases. These findings indicate that they require external tools to accurately disentangle and reconstruct those components. Nevertheless, as they succeed in identifying the presence of a component, we argue that LLMs are capable of deciding when to invoke an external decomposition tool, in line with the emerging paradigm of tool-using LLMs. Further implementation details and examples of the decomposition experiment, including prompt templates and evaluation settings, are provided in Appendix D.4.

**4.2.3 Index-Free vs. Index-Aware**   Existing studies [43, 89] have argued that longer input tokens may degrade LLM performance, and [26, 48] also prefer index-free prompts to keep inputs concise by omitting position indices. This motivates a direct comparison between index-free and index-aware prompts in the GT-provided setting. As shown in Table 4, LLMs perform worse with index-free prompts, recording significantly lower F1 scores. Index-aware prompts, despite roughly doubling the input token length, simplify the task by turning fragile position counting into direct retrieval, resulting in consistently higher localization accuracy. The input length influences the inference time, but the wall-clock latency is primarily determined by the number of output tokens. In the specific case of Qwen, index-free prompts occasionally resulted in verbose or rambling outputs (Appendix

Table 4: Performance comparison of TSAD with GT labels between index-free prompts (using only values) and index-aware prompts (using both indices and values).

| Backbone | Prompt | GT | Avg. Input Tokens | Avg. Output Tokens | Avg. Inference Time (s) | Standard Metric | | | Affiliation Metric | | |
|----------|--------|-----|-------------------|--------------------|-----------------------|------|--------|-----|------|--------|-----|
| | | | | | | Prec. | Recall | F1 | Prec. | Recall | F1 |
| Gemini-1.5 (Flash) | Baseline | ✗ | 4968 | 53.06 | **0.84** | 2.01 | 2.47 | 1.89 | 24.90 | 25.58 | 24.14 |
| | Index-free | ✓ | 5167 | 50.72 | 0.87 | 27.66 | 28.12 | 27.74 | 52.03 | 47.77 | 48.97 |
| | Index-aware | ✓ | 10058 | 50.83 | 1.20 | **64.79** | **62.46** | **62.83** | **67.88** | **67.54** | **67.39** |
| Qwen-2.5 (VL-72B-Inst.) | Baseline | ✗ | 4965 | 132.91 | 10.46 | 25.99 | 25.66 | 25.49 | 45.69 | 42.44 | 43.11 |
| | Index-free | ✓ | 5161 | 573.05 | 29.79 | 20.31 | 20.39 | 20.33 | 31.06 | 29.37 | 29.84 |
| | Index-aware | ✓ | 10051 | 59.01 | **6.45** | **60.44** | **64.47** | **60.16** | **73.71** | **72.87** | **72.57** |

Figure 19), which in turn substantially increased decoding length and inference time. Although we cannot pinpoint the exact cause, the lack of positional indices may have contributed by offering less concrete structure for the model to follow. Ultimately, including index tokens adds minor input overhead, but consistently leads to more dependable and accurate TSAD.

## 5 Discussion and limitation

In this section, we evaluate the practical value of using LLMs for TSAD. Although our simple method improves detection performance, it remains important to assess whether LLMs offer tangible advantages over conventional models. Previous works highlight two main benefits: (1) zero-shot capability without domain-specific training and (2) the ability to incorporate contextual information expressed in natural language. However, these advantages have not been extensively evaluated in comparison to non-LLM methods. we examine the extent to which current LLM-based methods deliver on these promises in real-world scenarios.

### 5.1 Comparison with conventional non-LLM TSAD methods

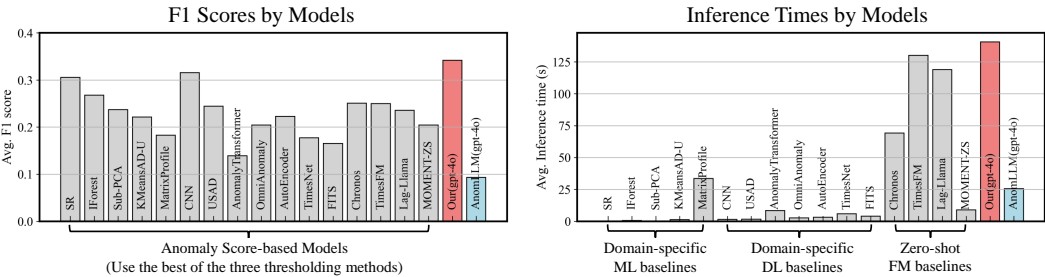

Figure 7: Comparison of F1 score and inference speed on the TSB-AD-U benchmark between 16 domain-specific ML/DL models and zero-shot FM/LLM-based models without any training.

We begin by comparing the accuracy and inference time of conventional AD methods and LLM-based methods using the TSB-AD-U benchmark. Unlike LLM-based methods, conventional methods typically produce an anomaly score for each timestamp, followed by a thresholding step to determine anomaly segments or binary label sequences. Since these methods are designed to compute scores that optimally separate anomalies from normal patterns, thresholding is treated as an orthogonal task. Consequently, evaluations are typically based on score-based metrics or the best F1 score obtained using GT labels.

To enable a fair comparison with LLM-based methods that directly predict anomaly labels, we adopt label-based evaluation metrics across all models. For conventional methods, we employ three widely used thresholding methods, such as percentile, MAD, and EVT-POT, and report the best-performing result among them. In terms of inference time, we measure the average inference time per sample for each method. To reduce the overall experimental cost, we limit the evaluation set by selecting time series from eight categories in the TSB-AD-U benchmark, focusing on those with relatively shorter lengths. Detailed model descriptions, full benchmark results, and dataset statistics are provided in Appendix C.3, Appendix Table 9, and Appendix B.2, respectively.

Figure 7 demonstrates that our LLM-based method achieves the highest average F1 score on the benchmark, indicating strong accuracy compared to non-LLM methods. While the method performs

reasonably well even in the absence of domain-specific training, its inference speed is significantly slower because of the overhead of decoding anomaly segments into textual descriptions. This trade-off underscores the promise of the method in low-resource or offline settings, while simultaneously revealing a key limitation in real-time scenarios.

## 5.2 Excluding known anomalies using context

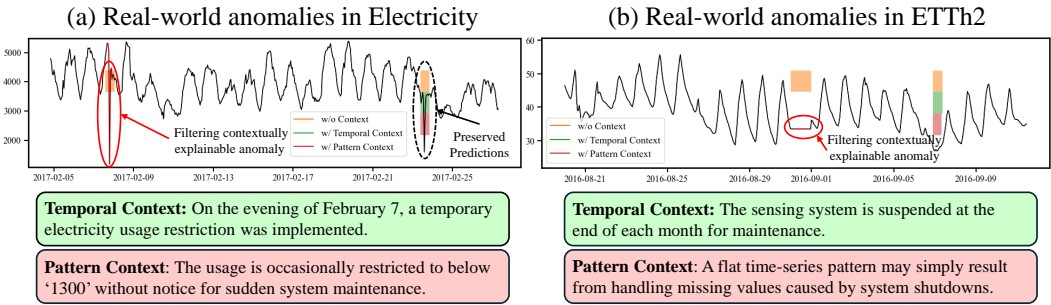

Figure 8: Qualitative results on two real-world time series, including anomalies commonly detected by conventional TSAD models and our (GPT-4o). The LLM incorporates sequential and contextual information to exclude explainable patterns while retaining other anomalies. The full prompt and the response are provided in Figure 20.

Another advantage of LLM-based methods over conventional models is their ability to use context through reasoning. We explore whether LLMs can perform context-aware anomaly filtering to remove uninteresting anomalies and to concentrate novel anomalies. Previous studies [45, 81] have focused mainly on detecting predefined anomaly types by using the context, but this is closer to pattern recognition than true anomaly detection. Also, real-world scenarios demand the discovery of previously unseen anomalies—those that deviate from normal behavior without prior examples. Using context to enhance the detection of known anomalies may bias the system toward those specific patterns, potentially reducing its sensitivity to unknown or novel failures. Thus, we pivot the use of contextual information in TSAD toward excluding known or uninteresting anomalies.

Figure 8 illustrates qualitative results on time series data commonly used in forecasting [1, 88], including anomalous patterns commonly detected by conventional models. We compare LLM predictions under different contexts. Without context, the LLM behaves similarly to conventional models, identifying abrupt dips or flat patterns as anomalies. However, when a temporal context and prior knowledge of expected time-series behavior are provided, the LLM can filter out explainable anomalies from its predictions. This demonstrates a distinctive strength of LLMs: the ability to integrate natural language context directly into inference, enabling interpretable context-aware filtering without rule-based systems or low-level code inspection.

Further implementation details, including the full prompt template (Figure 20) and the complete experimental setup for the context-aided TSAD analysis, are presented in Appendix D.6. We hope these findings inspire future research on LLM-based TSAD, particularly through the development of diverse datasets and evaluation benchmarks that reflect a wide range of real-world scenarios.

## 6 Conclusion

In this paper, we revisit the use of LLMs for TSAD by identifying two key challenges: temporal understanding and anomaly localization. Unlike prior work that focused solely on performance, we analyze these failure modes and propose a simple method, statistical decomposition with index-aware prompting, that improves detection accuracy without requiring additional training. The experimental results on benchmark datasets verify that our method consistently outperforms existing prompts and baseline models. We also discuss both a drawback of using LLMs in real-time settings: their latency compared to conventional TSAD methods, and a unique strength: context-aware filtering that better aligns with the semantic definition of anomalies. We believe our findings offer practical guidance on how and when to use LLMs effectively for TSAD.

## Acknowledgements

This work was supported by the Institute for Information & Communications Technology Planning & Evaluation (IITP) grant funded by the Korea government (MSIT) (RS-2019-II190075, Artificial Intelligence Graduate School Support Program (KAIST); RS-2024-00396828, Development of AI based low power 5G-A O-DU/O-CU), and the National Research Foundation of Korea (NRF) grant funded by the Korea government (MSIT) (No. RS-2025-00555621).

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

# A    Related Work

## A.1    Time-series Anomaly Detection

**Problem Formulation.** Let $\mathbf{x}_{1:T} = (x_1, \ldots, x_T) \in \mathbb{R}^{T \times d}$ be a time series sampled at a regular interval. Time-series anomaly detection seeks a function $f : \mathbb{R}^{T \times d} \to (\mathbf{s}_{1:T}, \mathbf{y}_{1:T})$ that, given the entire sequence, returns a vector of continuous anomaly scores $\mathbf{s}_{1:T} \in \mathbb{R}^T$ and, optionally, a binary label vector $\mathbf{y}_{1:T} \in \{0, 1\}^T$ obtained through a rule $y_t = \mathbb{I}[s_t > \tau]$. Each score $s_t$ quantifies how strongly the statistical properties of observation $x_t$ diverge from those expected under the normal generative process—after accounting for trend, seasonality and stochastic noise—while the label flags the corresponding time point (or minimal enclosing interval) as anomalous when the divergence exceeds a threshold $\tau$. The objective is thus to isolate individual timestamps or contiguous segments whose behavior departs significantly from the normal regime, signaling faults, rare events, or structural changes in the underlying system.

Within this formulation, we primarily consider two TSAD settings based on assumptions about label availability: unsupervised, where no label information is available beyond the input time series, and semi-supervised, where only data known to be normal is accessible. While certain existing methods [45, 89] incorporate anomaly information and thus deviate from these settings, we concentrate on the more realistic unsupervised and semi-supervised scenarios. Approaches in this area have evolved from classical statistical techniques to modern deep learning and foundation models.

**Statistical and Classical Models.** Traditional methods such as STL decomposition [17], Matrix Profile [84], and statistical thresholding [29] detect anomalies by explicitly modeling components like trend and seasonality. These approaches are efficient and interpretable but often struggle with non-stationary and multivariate data.

**Machine Learning-based Models.** Classical unsupervised models include Isolation Forest [44], Subspace PCA [2], and clustering-based methods [82]. While these algorithms are lightweight and interpretable, they often rely on handcrafted features and are limited in their ability to capture temporal dependencies.

**Deep Learning-based Models.** Deep models such as AutoEncoders [62], USAD [7], and Omni-Anomaly [65] reconstruct normal patterns to identify anomalies. More advanced architectures like the Anomaly Transformer [77] and TimesNet [75] leverage attention mechanisms to model long-term dependencies. These models achieve superior performance but require extensive training and tuning, and they often lack interpretability.

**Time-Series Foundation Models (TS-FMs).** Recent efforts like Chronos [6], TimesFM [18], MOMENT [24], and Lag-Llama [57] pretrain large models on diverse time-series datasets to enable zero-shot anomaly detection. These models show promise in generalization and transferability but still rely on scalar anomaly scores and thresholding for detection, limiting interpretability and precise localization.

**Comparison Across Models.** Table 5 summarizes key differences:

Table 5: Comparison of unsupervised TSAD model families in terms of modeling capacity, training requirements, and output type.

| Model Type | Temporal Modeling | Component Awareness | Domain Training | Output |
|---|---|---|---|---|
| Statistical | Low | Explicit | None | Score |
| ML-based | Moderate | Implicit | Light (generic heuristics) | Score |
| DL-based | High | Implicit | Required | Score |
| TS-FM | High | Learned | Pretrained | Score |
| LLM-based (ours) | Moderate | Structured (via prompt) | Pretrained | Label |

**Toward LLM-based TSAD.** In contrast, Large Language Models (LLMs) offer a novel paradigm for TSAD by generating structured, interpretable anomaly reports through prompt-based natural language reasoning [4, 20, 89]. Rather than producing scalar anomaly scores, LLMs can identify and explain anomalous intervals in plain text. However, existing LLM-based TSAD methods struggle

with precise localization and temporal abstraction, especially when relying on visual inputs [79] or unstructured token prompts [35, 43].

Our approach addresses these challenges by combining statistical preprocessing (e.g., de-seasonalization) with index-aware prompting to improve both detection and localization. This structured fusion of classical time-series principles and generative language reasoning establishes a new direction for interpretable and flexible unsupervised TSAD.

## A.2 LLM-based Time-series Anomaly Detection

Recent advances have explored the potential of using LLMs for TSAD, particularly in zero-shot or few-shot settings without domain-specific training. This line of research centers on a core question:

**Can LLMs detect time-series anomalies without training?** Studies [89, 20, 4, 79, 83, 45] have evaluated LLMs' ability to detect anomalies directly from raw or lightly preprocessed sequences via prompting. While some setups succeed in identifying simple anomalies, key limitations persist—namely, low detection accuracy, sensitivity to prompt variations, and hallucinations leading false positive cases. These investigations largely assess outcomes without examining the underlying causes of failure.

**Vision-based anomaly detection with MLLMs.** To bypass the difficulties LLMs face in capturing temporal patterns from textual input, some works [79, 89] propose converting time series into images and applying MLLMs to interpret the visualizations. This approach shows robustness to missing data and gross anomalies. However, it remains ineffective at detecting subtle irregularities hidden within complex, overlapping components. The transformation from sequence to image can also introduce distortions, particularly in high-dimensional settings, leading to inaccurate or unstable predictions.

**Context-aware and interpretable detection.** Other efforts [83, 45] investigate context-aware strategies that incorporate in-context learning, AnoCoT, and domain priors to improve both accuracy and interpretability. Although such methods enhance detection performance and support reasoning, they often rely on labeled anomaly examples during inference, which constrains their applicability to unsupervised or semi-supervised TSAD scenarios.

While earlier research has focused on whether LLMs can perform anomaly detection, our work shifts attention to understanding why they frequently fail. We identify two major barriers: (1) insufficient temporal abstraction and comprehension of latent components, and (2) structural misalignment between the input format and the task of localizing anomalies. Furthermore, we argue that visual representations alone are inadequate for precise localization, especially when positional indexing is essential. Instead, we advocate for structured textual representations that maintain explicit reference to index positions. Finally, unlike prior methods that leverage known anomalies to boost performance, we show that excluding such priors and instead using contextual cues better aligns with realistic, unsupervised TSAD conditions. All evaluations are conducted on the AnomLLM benchmark [89] to ensure consistency and fair comparison.

# B Benchmark Details

In this paper, we use two benchmark datasets. The first is the synthetic AnomLLM benchmark, and the second is the real-world TSB-AD benchmark dataset. In this section, we describe the characteristics and statistical properties of each dataset.

## B.1 AnomLLM benchmark

This appendix provides detailed information about the AnomLLM benchmark [89] datasets, which cover representative types of anomalies commonly discussed in time-series literature [16, 52]. The original dataset consists of 8 datasets in total: point anomalies, range anomalies, trend anomalies, frequency anomalies, noisy point anomalies, noisy trend anomalies, noisy frequency anomalies, and flat trend anomalies.

Among these, the first 4 are basic data forms, and the latter 4 are noisy variant forms. We only used the first 4 datasets, and we will provide an explanation of these datasets. As shown in the first three rows of Table 7, the four datasets differ in the presence or absence of the three components.

Table 6: Descriptions of the 21 prompt variants used in the AnomLLM benchmark. Prompts vary by supervision level (0-shot vs 1-shot), modality (text vs image), and auxiliary strategies such as CoT, statistical prefixing (PaP), arithmetic cues, and input formatting.

| Code | Variant | Description |
|---|---|---|
| A | 1shot-vision-cot | One-shot prompt with visual input and chain-of-thought (CoT) reasoning to guide step-by-step anomaly detection. |
| B | 1shot-vision-calc | One-shot visual input with a correct arithmetic example to test numeracy-based reasoning. |
| C | 1shot-vision-dyscalc | One-shot visual input with an incorrect arithmetic example to impair numeric reasoning. |
| D | 1shot-vision | Basic one-shot visual prompt without CoT or arithmetic guidance. |
| E | 0shot-vision-cot | Zero-shot visual input with CoT prompting to induce explicit anomaly reasoning. |
| F | 0shot-vision-calc | Zero-shot visual input with a correct arithmetic example included. |
| G | 0shot-vision-dyscalc | Zero-shot visual input with an incorrect arithmetic example to test robustness to misleading signals. |
| H | 0shot-vision | Basic zero-shot visual prompt with only the time series image. |
| I | 1shot-text-s0.3-cot | One-shot text input (subsampled to s0.3) with a CoT reasoning trace. |
| J | 1shot-text-s0.3 | One-shot prompt with s0.3 subsampled time series as plain text without reasoning. |
| K | 0shot-text-s0.3-tpd | Zero-shot text with Token-per-Digit formatting to aid digit-level modeling. |
| L | 0shot-text-s0.3-pap | Subsampled text with Prompt-as-Prefix: statistical summaries (mean, trend, etc.) precede the sequence. |
| M | 0shot-text-s0.3-dyscalc | Subsampled text with a misleading arithmetic example to degrade arithmetic reasoning. |
| N | 0shot-text-s0.3-csv | Text input formatted as CSV (index, value) to test structured data handling. |
| O | 0shot-text-s0.3-cot-tpd | Combines CoT prompting with tokenized digit input for fine-grained reasoning. |
| P | 0shot-text-s0.3-cot-pap | Combines CoT reasoning with Prompt-as-Prefix statistical context. |
| Q | 0shot-text-s0.3-cot-csv | Structured CSV input with CoT reasoning to examine logical behavior over tabular text. |
| R | 0shot-text-s0.3-cot | Basic CoT prompt with subsampled text input (no format enhancements). |
| S | 0shot-text-s0.3-calc | Subsampled text with a correct arithmetic example for numeracy-augmented detection. |
| T | 0shot-text-s0.3 | Plain zero-shot text prompt using s0.3-length series in space-separated format. |
| U | 0shot-text | Full-length zero-shot text prompt with raw values (space-separated). |

In addition, [89] specifies appropriate values for minimum anomaly duration, normal duration, and other parameters for each dataset. These details are presented in Table 7. Also, AnomLLM provided 21 prompt variants including CoT. The descriptions for each variant are in Table 6. And also, the results of experiments using these variants are in Table E.

### B.1.1 Point Anomalies.

Normal data consists of a periodic sine wave between -1 and 1. Anomalies present as noisy and unpredictable deviations from the normal periodic pattern, with frequency 0.03, normal duration rate 800.0, anomaly duration rate 30.0, minimum anomaly duration 5, minimum normal duration 200, and anomaly standard deviation 0.5. The example series is illustrated in Figure 9.

### B.1.2 Range Anomalies

Normal data comprises Gaussian noise with mean 0. Anomalies manifest as sudden spikes with values much further from 0 than the normal noise, with normal duration rate 800.0, anomaly duration rate 20.0, minimum anomaly duration 5, minimum normal duration 10, and anomaly size range (0.5, 0.8). The example series is illustrated in Figure 10.

Table 7: Comparison of anomaly statistics across different synthetic anomaly types.

| Statistic | Point | Range | Trend | Frequency |
|---|---|---|---|---|
| Trend component | ✗ | ✗ | ✓ | ✗ |
| Seasonality component | ✓ | ✗ | ✓ | ✓ |
| Noise component | ✗ | ✓ | ✗ | ✗ |
| # Time series | 400 | 400 | 400 | 400 |
| # Samples per time series | 1000 | 1000 | 1000 | 1000 |
| Minimum anomaly duration | 5 | 5 | 50 | 7 |
| Minimum normal duration | 200 | 10 | 800 | 20 |
| Average anomaly ratio | 0.0320 | 0.0236 | 0.0377 | 0.0341 |
| # Time series without anomalies | 117 (29.25%) | 121 (30.25%) | 230 (57.50%) | 40 (10.00%) |
| Average anomalies per series | 1.17 | 1.20 | 0.42 | 2.16 |
| Maximum anomalies per series | 4 | 5 | 1 | 7 |
| Average anomaly length | 27.26 | 19.73 | 88.61 | 15.77 |
| Maximum anomaly length | 165.0 | 113.0 | 200.0 | 111.0 |

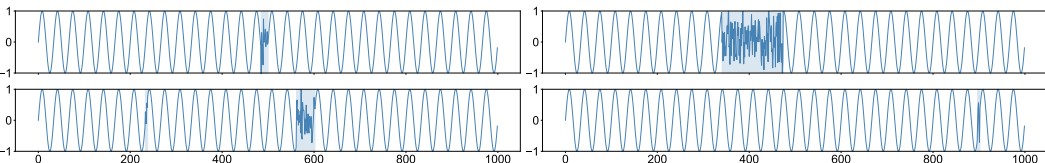

Figure 9: Example time series from the Point Anomalies dataset, with anomalies regions highlighted in blue.

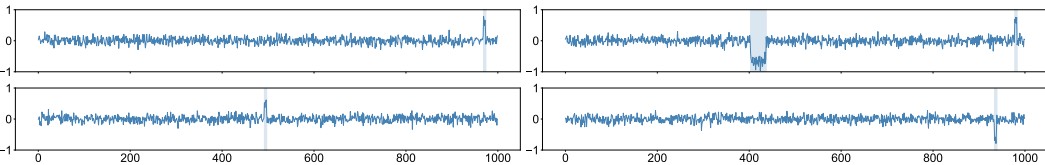

Figure 10: Example time series from the Range Anomalies dataset, with anomalies regions highlighted in blue.

### B.1.3 Trend Anomalies.

Normal data follows a steady but slowly increasing trend from -1 to 1. Anomalies appear as sections where the data increases much faster or decreases, deviating from the normal trend, with trend negation probability 50%, frequency 0.02, normal duration rate 1700.0, anomaly duration rate 100.0, minimum anomaly duration 50, minimum normal duration 800, normal slope 3.0, and abnormal slope range (6.0, 20.0). The example series is illustrated in Figure 11.

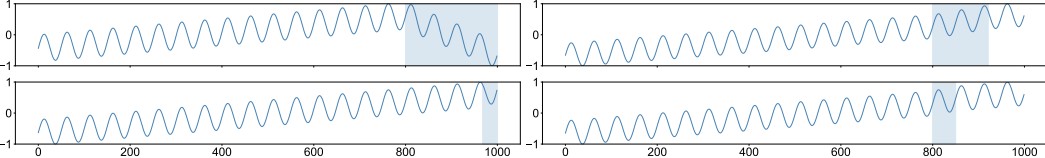

Figure 11: Example time series from the Trend Anomalies dataset, with anomalies regions highlighted in blue.

### B.1.4 Frequency Anomalies.

Normal data is characterized by a periodic sine wave between -1 and 1. Anomalies occur as sudden changes in frequency, producing irregular periods between peaks, with frequency 0.03, normal

duration rate 450.0, anomaly duration rate 15.0, minimum anomaly duration 7, minimum normal duration 20, and frequency multiplier 3.0. The example series is illustrated in Figure 12.

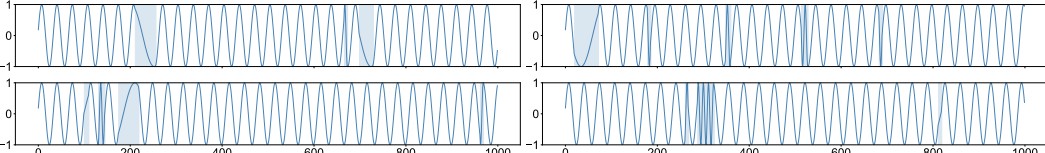

Figure 12: Example time series from the Frequency Anomalies dataset, with anomalies regions highlighted in blue.

## B.2 TSB-AD Benchmark

This section describes the real-world dataset, TSB-AD benchmark [46], used in Section 4. The benchmark initially collected 13 univariate and 20 multivariate datasets. Following the curation process within TSB-AD, they obtained a total of 23 univariate and 17 multivariate datasets. Internally, datasets with a dimension of 1 were referred to as TSB-AD-U, among which we used the evaluation subset for our experiments.

To reduce the overall experimental cost, we restrict the evaluation set by selecting time series from eight categories (highlighted in green in Table 8) within the TSB-AD-U benchmark, focusing on those with relatively shorter lengths.

Table 8: TSB-AD-U evaluation benchmark dataset statistics. The total length is computed as the product of the average time series length and the number of series (Count).

| Dataset | Count | Dim | Total Len | Avg. # Anomaly | Avg. Len | Anomaly Ratio | Category |
|---|---|---|---|---|---|---|---|
| NEK [64] | 8 | 1 | **8,584** | 2.9 | 51.1 | 8.0% | P&Seq |
| TAO [50] | 2 | 1 | **20,000** | 838.7 | 1.1 | 9.4% | P&Seq |
| MSL [32] | 7 | 1 | **23,111** | 1.3 | 130.0 | 5.8% | Seq |
| Power [36] | 1 | 1 | **35,040** | 4.0 | 750.0 | 8.5% | Seq |
| Daphnet [8] | 1 | 1 | **38,774** | 6.0 | 384.3 | 5.9% | Seq |
| YAHOO [39] | 30 | 1 | **45,270** | 5.5 | 2.5 | 0.6% | P&Seq |
| SED [11] | 2 | 1 | **59,998** | 14.7 | 64.0 | 4.1% | Seq |
| TODS [38] | 13 | 1 | **65,000** | 97.3 | 18.7 | 6.3% | P&Seq |
| NAB [3] | 23 | 1 | 114,758 | 1.6 | 370.1 | 10.6% | Seq |
| Stock [72] | 8 | 1 | 130,000 | 1246.9 | 1.1 | 9.4% | P&Seq |
| SMAP [32] | 17 | 1 | 133,770 | 1.2 | 210.1 | 2.8% | Seq |
| CATSv2 [21] | 1 | 1 | 300,000 | 19.0 | 778.9 | 4.9% | Seq |
| WSD [86] | 20 | 1 | 348,639 | 5.1 | 25.4 | 0.6% | Seq |
| SWaT [47] | 1 | 1 | 419,919 | 27.0 | 1876.0 | 12.1% | Seq |
| OPP [61] | 27 | 1 | 449,396 | 1.4 | 653.4 | 6.4% | Seq |
| MGAB [71] | 8 | 1 | 780,000 | 9.7 | 20.0 | 0.2% | Seq |
| SMD [65] | 33 | 1 | 791,393 | 2.4 | 173.7 | 2.0% | Seq |
| LTDB [23] | 8 | 1 | 800,000 | 127.5 | 144.5 | 18.6% | Seq |
| IOPS [33] | 15 | 1 | 1,048,682 | 25.6 | 48.7 | 1.3% | Seq |
| Exathlon [34] | 30 | 1 | 1,324,295 | 3.1 | 1577.3 | 11.0% | Seq |
| SVDB [25] | 20 | 1 | 3,016,800 | 36.4 | 292.5 | 3.6% | Seq |
| UCR [76] | 70 | 1 | 3,806,932 | 1.0 | 198.9 | 0.6% | P&Seq |
| MITDB [23] | 7 | 1 | 4,400,000 | 68.7 | 451.9 | 4.2% | Seq |

## C  Model details

### C.1  LLMs

To evaluate whether our proposed strategies generalize across different LLMs, we conduct experiments using two open-source LLMs and two commercial API-based LLMs. For the open-source models, we employ InternVL2-Llama3-76B and Qwen2.5-VL-72B-Instruct. For the API-based models, we use Gemini-1.5-Flash by Google and GPT-4o by OpenAI.

The open-source models are hosted on an A100 4-GPU machine using the lmdeploy library, and queries are issued locally through this setup. For Gemini-1.5-Flash and GPT-4o, we directly send requests to their respective APIs without using multi-threading, in order to accurately measure per-query latency.

**Intern-VLM** [15, 14], also known as InternVL2, is an open-source multimodal LLM designed to close the performance gap between open and commercial models in multimodal understanding. It combines a powerful vision encoder, InternViT, with support for dynamic high-resolution processing (up to 4K), and a bilingual training corpus. The language component is initialized with Hermes-2-Theta-LLaMA3-70B. InternVL2 achieves state-of-the-art results on 8 out of 18 benchmarks, outperforming some proprietary models in tasks like chart understanding.

In the AnomLLM experiments, performance with and without visual input was assessed using a trivial image. The model achieved an MMLU-Pro [73] score of 52.95 without the image and 53.26 with the image. This suggests that visual input does not negatively affect language performance. It is also noted that the slightly lower MMLU score is attributable to the Hermes-based language backbone, which underperforms compared to Meta's official LLaMA3-70B-Instruct.

**Qwen** [9], specifically Qwen2.5-VL-72B-Instruct, is a large-scale multimodal language model developed by Alibaba Group. It builds upon the Qwen2.5 architecture and integrates a visual encoder to support image-text understanding tasks. The model features a 72-billion parameter transformer-based language backbone and leverages a high-resolution vision module. Pretrained on a mixture of web-scale bilingual corpora and diverse vision-language data, Qwen2.5-VL is optimized for instruction-following scenarios with multimodal inputs.

Qwen2.5-VL-72B-Instruct demonstrated strong reasoning capabilities, maintaining consistent performance both with and without visual input. For example, it achieved an MMLU-Pro score of 71.2 [10], compared to GPT-4o's 72.6. On the MathVision_FULL benchmark, Qwen scored 38.1 while GPT-4o scored 30.4—suggesting a slight advantage for Qwen when leveraging multimodal context. These results highlight the robustness of Qwen's visual-language integration and training pipeline.

**Gemini-1.5-Flash** [70] is a proprietary multimodal language model developed by Google, optimized for high-throughput and cost-efficient inference. It supports long-context processing and performs well across a wide range of multimodal tasks such as image understanding, classification, summarization, and content generation from visual, audio, or video inputs. Despite its speed-oriented design, it maintains competitive quality comparable to other Gemini Pro models while significantly reducing operational cost.

In the AnomLLM study, a small white image (10×10 pixels) was added to text prompts to test whether including visual inputs would affect model performance. The MMLU-Pro score remained stable—59.12 without image and 59.23 with image—indicating that the same model backend is likely used regardless of input modality, and that vision input does not degrade text reasoning capabilities.

**GPT-4o** [51], specifically the `gpt-4o-2024-08-06` version, is OpenAI's latest flagship multimodal model that unifies vision and text understanding. Unlike earlier versions that relied on separate encoders for different modalities, GPT-4o adopts a fully integrated transformer capable of jointly processing and reasoning across modalities with native support.

In the AnomLLM experiments, GPT-4o showed robust performance on both textual and multimodal benchmarks. Its MMLU-Pro score was 72.6 and GPQA score was 53.6. GPT-4o stands out for its balanced strength across diverse tasks and its ability to seamlessly integrate visual context without requiring explicit prompting strategies or auxiliary modules.

## C.2  Our strategies

**Statistical time-series decomposition.** To compare with LLM-based TSAD approaches, we conducted experiments using conventional statistical decomposition techniques. Specifically, we employed the `seasonal_decompose` function from the `statsmodels` package, which decomposes a univariate time series into three components: trend, seasonality, and residual. This method assumes that the observed signal consists of systematic patterns combined with random noise, and supports both additive and multiplicative models. Following prior work [46], we adopt the additive model, defined as:

$$Y[t] = T[t] + S[t] + e[t],$$

where $Y[t]$ is the observed value at time $t$, $T[t]$ is the trend component, $S[t]$ is the seasonal component, and $e[t]$ is the residual.

The decomposition process uses moving averages to isolate the seasonal component, with the period either inferred from the time index or specified manually. When `two_sided=True`, a centered moving average is used; when `False`, the filter is causal, relying only on past values.

This method yields a structured separation of the time series into long-term trends, recurring seasonal patterns, and residuals. Anomalies are detected by analyzing the residuals, which capture deviations unexplained by the expected structure.

**Our prompts.** In our experiments, we conducted tests using both 0shot-Text and 0shot-Text-Vision while providing the corresponding indexes. We used the prompt shown in Figure 13 for these experiments.

---

**Our (0shot-Text-Vision)**

---

I will provide you with time-series data recorded at hourly intervals, along with a plotted time-series image.
Here is time-series data in (index, value) format:
<history>
(index, value)
(index, value)
(index, value)
...
</history>
Assume there are up to 5 anomalies.

Detect ranges of anomalies in this time series, in terms of the index of time-series data, considering the plotted image.
List one by one, in JSON format.
If there are no anomalies, answer with an empty list []. Do not say anything other than the answer.

---

**Response**: [(start, end), (start, end), ...]

---

**Our (0shot-Text)**

---

I will provide you with time-series data recorded at regular intervals.
Here is time-series data in (index, value) format:
<history>
(index, value)
(index, value)
(index, value)
...
</history>
Assume there are up to 5 anomalies.

Detect ranges of anomalies in this time series, in terms of the index of time-series data.
List one by one, in JSON format.
If there are no anomalies, answer with an empty list []. Do not say anything other than the answer.

---

**Response**: [(start, end), (start, end), ...]

---

Figure 13: The prompts used in the experiments: 0shot Text-Vision and 0shot Text.

### C.3 Conventional non-LLM-based TSAD Methods in Section 5.1

**(i) ML baseline**

- **SR** [59] first applies the Fourier Transform to the input data and then computes the spectral residual from the log-amplitude spectrum. This modified spectrum is converted back to the time domain using the Inverse Fourier Transform, resulting in a saliency map. Anomaly scores are derived by measuring how much each value in the saliency map deviates from its moving average.

- **IForest** [44] builds a binary tree where the length of the path from the root to a given node indicates the likelihood of an anomaly—shorter paths imply a higher chance that the point is anomalous.

- **Sub-PCA** [2] projects subsequences into a lower-dimensional subspace, and large deviations from this projection space are treated as anomalies due to the breakdown of linear assumptions.

- **KMeans-AD** [82] assigns each subsequence to the nearest cluster centroid using k-means and then calculates anomaly scores as the distance between the subsequence and its assigned centroid.

- **MatrixProfile** [84] detects anomalies by identifying subsequences with unusually large distances to their nearest neighbors, thus uncovering patterns that differ from the rest of the time series.

**(ii) DL baseline**

- **CNN** [49] trains a convolutional network to predict future time steps from recent observations, and the anomaly score is derived from the prediction error at each point.

- **USAD** [7] combines reconstruction and adversarial loss from two autoencoders, where discrepancies between input and reconstruction, amplified by a discriminator, yield the anomaly score.

- **AnomalyTransformer** [77] introduces an Anomaly-Attention mechanism that captures temporal dependencies and assigns anomaly scores based on deviations in attention-based associations.

- **OmniAnomaly** [65] uses a stochastic recurrent neural network with variational inference and normalizing flows to learn representations of normal patterns, identifying anomalies via reconstruction probabilities.

- **AutoEncoder** [62] projects vector to the lower-dimensional latent space and reconstruct it through the encoding-decoding phase, where anomalies are typically characterized by evident reconstruction deviations.

- **TimesNet** [75] adopts a general-purpose block for multivariate time series that adaptively learns multiple periodicities via a hierarchical temporal convolutional structure.

- **FITS** [80] interpolates time series into the complex frequency domain to detect anomalies, leveraging the efficiency of frequency-domain manipulation with minimal parameters.

**(iii) Foundation Model-based Method**

- **Chronos** [6] tokenizes time series values through scaling and quantization into discrete tokens and trains a T5 model using a standard cross-entropy loss for anomaly detection.

- **TimesFM** [18] pretrains a decoder-style attention model using input patching on a large-scale time series corpus to learn general representations for various forecasting and detection tasks.

- **Lag-Llama** [57] introduces a decoder-only transformer architecture that conditions on lagged inputs, making it suitable for probabilistic forecasting and anomaly detection on univariate time series.

- **MOMENT** [24] is a T5-based encoder model pre-trained using masked time-series modeling to reconstruct masked values, enabling fine-tuning for downstream anomaly detection tasks.

**(iv) Thresholding Method**

- **Precentile** [6] selects a fixed quantile (e.g., top 5%) of the anomaly scores as the threshold to label outliers, assuming the score distribution reflects normal vs. abnormal points.

- **MAD** [29] computes the median absolute deviation to identify outliers, offering robustness against extreme values and non-Gaussian distributions.
- **EVT-POT** [63, 67] fits the tail of the anomaly score distribution with the Generalized Pareto Distribution via Peaks Over Threshold, enabling dynamic anomaly thresholding.

# D  Experiment details

## D.1  Understanding experiments in Section 2.1

---

**Determining whether anomaly exists**

---

Determine whether this time series contains any anomalies.
Respond with exactly 'True' if anomalies are present, or 'False' otherwise.
Output only 'True' or 'False'.

---

**Response**: 'True' or 'False'

---

Figure 14: A prompt that determines the presence of anomalies and outputs a binary result.

To evaluate how well LLMs understand different types of anomalies in the TSAD setting, we design an experiment using datasets that cover four representative anomaly types: trend shift, frequency change, context-deviating point, and out-of-range anomalies, as proposed in AnomLLM. The dataset consists of 1,600 time series samples, with 400 instances per anomaly type. We convert the original interval-based labels into binary labels by marking whether any anomaly interval is present in a given time series, effectively framing the task as instance-level TSAD without requiring localization. Accordingly, the prompt output format is modified from interval prediction to a binary decision, as illustrated in Figure 14.

The experimental setup follows the same configuration as the TSAD task in AnomLLM, with the only modification being the binary output format. We adopt the F1-Macro score to fairly assess performance across both normal and anomalous classes. As a lower-bound reference, we include a constant classifier baseline to delineate the threshold for meaningful predictions. The experiment leverages the 21 prompt variants proposed in AnomLLM, with detailed descriptions provided in Table 6, and full dataset statistics presented in Table 7.

## D.2  Localization experiments in Section 2.2

---

**A 0shot-Vision prompt for image sequences with ground truth labels**

---

Detect ranges of anomalies, highlighted by blue-colored regions, in this time series, in terms of the x-axis coordinate.

List one by one, in JSON format. If there are no anomalies, answer with an empty list [].

Output template:
[{ "start": number , "end": number } , ...]

---

**Response**: [{ "start": number , "end": number } , ...]

---

Figure 15: The 0shot-vision prompt for localization-only experiment.

This section provides detailed information about the experiment designed to evaluate how well the model localizes anomalies during TSAD using AnomLLM. The dataset and evaluation protocol follow the AnomLLM benchmark. To minimize performance degradation caused by failures in understanding, we explicitly provide the ground-truth anomaly segments during the detection process for each sample as shown in Figure 9.

Specifically, this experiment uses the 0-shot vision-based prompt, which demonstrated strong performance in the original AnomLLM benchmark. When presenting the image, the ground-truth

anomaly segments are visually highlighted within the image to clearly indicate the anomalous regions. TSAD is then performed using a prompt (Figure 15) that explicitly instructs the model to utilize this information.

## D.3 Counting experiments in Section 3.2

---

**Counting prompt**

---

Below is a numerical sequence:
{selected_series}

Count the total number of elements in this sequence.
Please return only the count as a single integer without any additional text or explanation.

---

**Response**: integer

---

Figure 16: The prompt used in the experiment to count elements in a given sequence.

This section offers additional details on the experiment introduced earlier. First, we conduct a sequence-length counting task by randomly sampling 100 sequences from the AnomLLM benchmark. Sequence lengths range from 1 to 1,000 and are binned into ten 100-step intervals (1–100, 101–200, ..., 901–1,000), with exactly ten sequences drawn from each interval to ensure a uniform distribution. We test four large language models and four different separator tokens used to delimit values within a sequence. Each model is instructed to return a single integer representing the sequence length. As shown in Figure 16, The prompt template is identical to the one adopted in 0shot-text variant of AnomLLM, except that the desired output is replaced with the sequence length.

## D.4 Time-series decomposition experiments in Section 4.2

This section provides further details on our evaluation of LLMs' ability to perform time-series decomposition without relying on external tools. We consider two subtasks: (1) identifying whether a time-series contains trend and/or seasonality components, and (2) generating the corresponding component sequences.

For the detection task, we constructed prompts that directly asked whether the given time-series contains trend or seasonal structure. Example prompts are provided in Figure 17. We used synthetic sequences with known decomposition structure sampled from AnomLLM as ground truth. As a baseline, we apply statistical decomposition and use a threshold to determine whether each component is present in a time series. The algorithm is shown in Algorithm 1. Specifically, a component is considered present if its amplitude exceeds a certain percentage of the original series' value range. For example, with a threshold set to 10%, we compute the range of the original time series (i.e., max minus min), and if the amplitude of the seasonal component exceeds 10% of this range, we classify the series as containing seasonality. This threshold-based rule transforms the continuous decomposition output into a binary detection result. A comparison of F1 scores across models is presented in Figure 6(a)

Specifically, we used the following hyperparameters.

- thresh_trend=0.57

- thresh_seasonal=0.1

- thresh_resid=0.15

For the generation task, models were asked to output the trend and seasonality as separate numerical sequences given the original input. Prompt examples for this task are shown in Figure 17. While some responses preserved global shape, most models failed to accurately reproduce the true components. In particular, errors increased with sequence length, and artifacts such as baseline drift and amplitude attenuation were frequently observed. A comparison of mean absolute errors across models is presented in Figure 6(b), and representative examples are illustrated in Figure 6(c).

---

**Component detection prompt**

---

Time series data can typically be decomposed into three main components such as Trend, Seasonality, and Residuals.
In this analysis, Residuals are further divided into Noise and Anomalies.

The four components exhibit the following characteristics:
 Trend: A long-term directional pattern or consistent upwarddownward movement.
 Seasonality: Regular and predictable cycles repeating at consistent intervals.
 Residuals: The remaining noise or anomalies in the data.

Below is the given time series data:
{time_series}

Analyze the provided time series carefully.
Determine if each of these components (Trend, Seasonality, Residuals) is present (1) or absent (0).

Provide your answer strictly in the following JSON format:
{"Trend":       0    or    1,       "Seasonality":       0    or    1,       "Residuals":       0    or    1}

---

**Response**: {"Trend": 0 or 1, "Seasonality": 0 or 1, "Residuals": 0 or 1}

---

---

**Component generation prompt**

---

Time series data can typically be decomposed into three main components: Trend, Seasonality, and Residual.

The three components exhibit the following characteristics:  Trend: A long-term directional pattern or consistent upward/downward movement.
 Seasonality: Regular and predictable cycles repeating at consistent intervals.
 Residual: The remaining noise or anomalies in the data.

Below is the given time series data:
{sampled_series}

CRITICAL INSTRUCTION: Extract ONLY the trend component from this time series.

EXACT LENGTH REQUIREMENT: Your output MUST contain EXACTLY {num_samples} values  one trend value for EACH value in the original time series.

DECIMAL PRECISION: Maintain the same level of precision as the original data (2 decimal places). Example: If original values are like "-0.63", "-0.57", trend values should also have 2 decimal places like "-0.62", "-0.61".

Count the values in the original data carefully. The original has exactly {num_samples} values, so your trend component must also have exactly num_samples values, no more and no less.

Return your answer of {component} as space-separated numbers in the following JSON format:
{
"{component}": "value1 value2 value3 ... value{num_samples}"
}

DO NOT abbreviate or shorten the output. Include EVERY single trend value with 2 decimal places.

---

**Response**: "{component}": "value1 value2 value3 ... value{num_samples}"

---

Figure 17: Component detection and generation prompts used in the decomposition experiment.

## D.5  Index-free vs. Index-aware experiments in Section 4.2

Prior works have often favored the index-free approach, as it requires only a simple list of values that compose the sequence, resulting in a short and concise format. This simplicity can make it easier for LLMs to understand the sequence, potentially contributing positively to task performance. However,

**Algorithm 1** Component Detection in Time Series

---

**Require:** Time series sequence, seasonality period, Threshold values $thresh_{trend}$, $thresh_{seasonal}$, $thresh_{resid}$

**Ensure:** Binary indicators for trend, seasonal, and residual components

1: **function** DETECTCOMPONENTS($seq$, $period$, $thresh_{trend}$, $thresh_{seasonal}$, $thresh_{resid}$)
2:      $decomposition \leftarrow$ SeasonalDecompose($seq$, model $=$ "additive", period $= period$)
3:      $tr \leftarrow decomposition.trend$
4:      $se \leftarrow decomposition.seasonal$
5:      $re \leftarrow decomposition.resid$
6:      $p_{tr} \leftarrow \mathbf{1}(\max(|tr|) \geq thresh_{trend})$         ▷ Binary indicator for trend
7:      $p_{se} \leftarrow \mathbf{1}(\max(|se|) \geq thresh_{seasonal})$     ▷ Binary indicator for seasonal
8:      $p_{re} \leftarrow \mathbf{1}(\max(|re|) \geq thresh_{resid})$       ▷ Binary indicator for residual
9:      **return** $p_{tr}, p_{se}, p_{re}$
10: **end function**

---

---

**TSAD with GT labels with text sequence only (Index-aware)**

---

... (968,-0.03), (969,-0.1), (970,0.12), (971,0.1), (972,0.24), (973,0.11), (974,0.03), (975,-0.02), (976,0.09), (977,0.14), (978,-0.03), (979,-0.01), (980,-0.18), (981,0.06), (982,-0.0), (983,0.14), (984,0.74), (985,0.73), (986,0.78), (987,0.56), (988,0.98), (989,0.78), (990,0.7), (991,0.8), (992,0.64), (993,0.82), (994,0.69), (995,0.55), (996,0.51), (997,0.9), (998,0.78), (999,0.58)

Assume there are up to 5 anomalies. Detect the ranges of anomalies in this time series based on the x-axis coordinate, where the x-axis coordinate refers to the x value in (x, y) data. The patterns of anomalies are as follows.
anomaly data 1: 0.07 -0.4 -0.71 -0.8 -0.58 -0.46

anomaly data 2: 0.12 0.46 0.75 0.7 0.82 0.65

anomaly data 3: 0.14 0.74 0.73 0.78 0.56 0.98 0.78 0.7 0.8 0.64 0.82 0.69 0.55 0.51 0.9 0.78 0.58
List one by one, in JSON format. If there are no anomalies, answer with an empty list [].

Output template: [{"start": ..., "end": ...}, {"start": ..., "end": ...}...]

---

**Response**: json [ {"start": 414, "end": 460}, {"start": 215, "end": 221}, {"start": 984, "end": 999} ]

---

Figure 18: A successful case in querying index-aware prompts for text-only TSAD with ground-truth labels.

in TSAD, the absence of index information introduces a significant drawback: when models are required to output anomaly intervals, they must internally perform a counting task to determine positions.

In contrast, the index-aware approach explicitly includes index–value pairs, requiring additional separator tokens and leading to longer and more complex input formats. This added complexity can potentially degrade task accuracy, especially when the context window becomes too long.

To evaluate the trade-offs between these two approaches, we conducted an experiment designed to isolate the effect of index formatting. To control for other variables, we only used text-based sequences without any accompanying image sequences. Similar to the localization experiment in Section 2.2, we provided ground-truth anomaly labels and compared localization performance under two conditions: with and without index information.

When using only text sequences, we included the ground-truth anomaly segments in the prompt and designed the task such that the model had to identify these anomalies in the sequence as shown in Figure 18.

As a result, consistent with our earlier discussion, we found that even though the index-aware prompts were longer and more complex, they outperformed index-free prompts. The benefit of eliminating the need for internal counting outweighed the cost of increased input length. In the case of Qwen,

**Unnecessary code generation and reasoning outputs when using index-free prompts.**

**Response:** To detect anomalies, we compute the mean (μ) and standard deviation (σ) of the series, and declare any point more than 3σ away from μ an anomaly.

**Steps:**

1. Compute the mean (μ).

2. Compute the standard deviation (σ).

3. Mark points further than 3σ from μ as anomalies.

Let's calculate these values and identify the anomalies.

### Calculations:

1. **Mean (μ)**: The average value of the dataset.
2. **Standard Deviation (σ)**: A measure of the amount of variation or dispersion of a set of values.

```python
import numpy as np

data = [0.05, -0.02, -0.06, -0.01, 0.14, 0.12, -0.04, 0.05, -0.04, 0.1, 0.03, 0.02, 0.1, -0.0, 0.1, -0.06, -0.01, 0.09, -0.1, ...

mean = np.mean(data)
std_dev = np.std(data)
anomaly_threshold = 3 * std_dev

anomalies = []
for i, value in enumerate(data):
    if abs(value - mean) > anomaly_threshold:
        anomalies.append(i)

# Grouping anomalies into ranges
anomaly_ranges = []
if anomalies:
    start = anomalies[0]
    for i in range(1, len(anomalies)):
        if anomalies[i] != anomalies[i - 1] + 1:
            anomaly_ranges.append("start": start, "end": anomalies[i - 1])
            start = anomalies[i]
    anomaly_ranges.append({"start": start, "end": anomalies[-1]})

print(anomaly_ranges)
```
### Output: ```json [ {"start": 18, "end": 18}, {"start": 25, "end": 25}, {"start": 27, "end": 27}, ...

Figure 19: A failure case in querying index-free prompts for text-only TSAD with ground-truth labels. The quantitative evaluation results of the index-free and index-aware prompts are presented in Section 4.2.

index-free prompts occasionally resulted in the generation of unnecessary code or reasoning steps (Figure 19), suggesting that the inherent difficulty of the task without index cues may be responsible for such verbose outputs.

## D.6 Context-aided TSAD experiments in Section 5.2

---

**Prompt and Response of ETTh2 example**

---

I will provide you with time-series data recorded at hourly intervals, along with a plotted time-series image.
Here is time-series data in (timestamp, value) format:
<history>
(2016-08-20 01:00:00, 46)
(2016-08-20 02:00:00, 45)
(2016-08-20 03:00:00, 44)
...
(2016-09-11 08:00:00, 34)
(2016-09-11 09:00:00, 35)
</history>

Here is context to understand the time series.

The following event occurred in given period. A time-series pattern that can be explained by this event not considered an anomaly.
* A flat time-series pattern may simply result from handling missing values caused by system shutdowns.

Assume there are up to 3 anomalies.

Detect ranges of anomalies, which are not explained by context in this time series, in terms of the timestamp of time-series data, considering the plotted image.

Step1) Present possible anomalies with reason.

Step2) Remove explainable anomalies by context.

Step3) List one by one, in JSON format. If there are no anomalies, answer with an empty list []. Do not say anything other than the answer.
Output template:
[{"start timestamp": ..., "end timestamp": ...}, {"start timestamp": ..., "end timestamp": ...}...]

Please answer to all steps.

---

**Response**:
Step1) Present possible anomalies with reason.

1. From 2016-08-31 00:00:00 to 2016-08-31 23:00:00, the time series shows a flat pattern at a value of 33, which could be considered an anomaly due to its deviation from the usual pattern of fluctuations.
2. From 2016-09-06 22:00:00 to 2016-09-07 08:00:00, the time series shows a sudden drop to a lower value range, which could be considered an anomaly due to its deviation from the usual pattern.

Step2) Remove explainable anomalies by context.

1. The flat pattern from 2016-08-31 00:00:00 to 2016-08-31 23:00:00 can be explained by the context provided, which mentions that flat patterns may result from handling missing values due to system shutdowns. Therefore, this is not considered an anomaly.
2. The sudden drop from 2016-09-06 22:00:00 to 2016-09-07 08:00:00 does not have an explanation provided by the context, so it remains a potential anomaly.

Step3) List one by one, in JSON format.

```json [{"start timestamp": "2016-09-06 22:00:00", "end timestamp": "2016-09-07 08:00:00"}] ```

Figure 20: The prompt and response of qualitative result on (b) potential range anomalies in ETTh2.

Conventional TSAD has relied heavily on numerical sequences, using statistical or neural network-based models. While there have been attempts to incorporate textual information into these models, such approaches often depend on synthetic datasets due to the difficulty of collecting paired text

and time-series data. As a result, they fall short in terms of generality and reasoning capabilities compared to LLMs. Thus, without other modality, these conventional models generally depend on an assumption in which anomalies are defined by sparsity or abrupt changes in the time-series patterns.

However, in real-world applications, not all sparse or abrupt patterns are of actual concern. For instance, in e-commerce platforms, promotional events can cause a sudden surge in product sales. Traditional models may flag such a change as an anomaly, despite it being an expected and well-known outcome. While it is possible to exclude known time windows using rule-based filters, this quickly becomes labor-intensive and inflexible when many events or patterns need to be handled. In contrast, simply describing these contexts in natural language would be a far more scalable and user-friendly solution.

In this situation, the reasoning ability of LLMs to interpret natural language context offers a valuable control interface for TSAD. To explore this capability, we conducted a study using real-world datasets—ETTh2 and Electricity. We first ran eight conventional anomaly detection models on these datasets and extracted time segments that were commonly detected as anomalies. We then generated domain-informed contextual descriptions for each of these segments, simulating plausible scenarios such as a known event causing a sudden drop, or a specific pattern being aligned with external factors. These contexts provided time-related or value-specific cues in natural language.

Using the prompt as shown in Figure 20, we evaluated the anomaly decisions of the LLM both with and without these contextual inputs. Our results show that the LLM could effectively integrate temporal cues, numerical patterns, and contextual knowledge to suppress false positives—i.e., anomalies detected by conventional models that users would not consider problematic.

Unlike conventional models that rigidly detect anomalies based on pattern shifts alone, LLM-based context-aided TSAD can reinterpret these shifts in light of the user's intent and domain knowledge. This allows for "justifiable detection," where the LLM not only detects changes but reasons whether those changes are relevant anomalies given the context. The LLM thus transforms anomaly detection into an interactive, user-aligned process that prioritizes interpretability and flexibility.

This paradigm shift points to several promising directions for future work: improving generalization across domains, ensuring real-time applicability of context-aware detection, and defining minimal requirements for effective contextual prompts. Notably, because the meaning of "anomaly" can vary depending on the user's objective or operational context, LLMs open the door to a new class of objective-aware anomaly detection systems—systems that adapt dynamically to changing definitions of what matters, rather than adhering to a fixed statistical criterion.

Table 9: TSAD results on the evaluation set of TSB-AD-U benchmark. The summarized results are presented in Section 5.1.

| Method | Thresholding | Standard Metrics | | | Affiliation Metrics | | | Inference Time |
|---|---|---|---|---|---|---|---|---|
| | | Prec. | Recall | F1 | Prec. | Recall | F1 | |
| AnomLLM (GPT-4o) | – | 12.79 | 11.96 | 9.32 | 39.59 | 30.54 | 31.59 | 25.52 |
| AnomalyTransformer | Percentile | 15.21 | 40.35 | 13.91 | 54.04 | 86.67 | 64.13 | 8.49 |
| FITS | MAD | 13.88 | 38.54 | 16.54 | 56.03 | 86.48 | 65.73 | 4.05 |
| TimesNet | MAD | 14.40 | 39.63 | 17.71 | 56.87 | 85.48 | 66.02 | 5.91 |
| MatrixProfile | MAD | 19.73 | 28.82 | 18.29 | 61.87 | 59.30 | 54.80 | 33.58 |
| MOMENT-ZS | Percentile | 26.07 | 27.35 | 20.42 | 65.33 | 71.04 | 63.82 | 8.90 |
| OmniAnomaly | MAD | 27.91 | 30.81 | 20.46 | 63.14 | 83.35 | 68.08 | 2.66 |
| KMeansAD-U | Percentile | 31.58 | 25.28 | 22.14 | 66.57 | 50.01 | 50.77 | 1.28 |
| AutoEncoder | MAD | 27.12 | 27.44 | 22.27 | 66.21 | 60.82 | 57.25 | 3.16 |
| Lag-Llama | MAD | 24.17 | 45.67 | 23.54 | 65.20 | 92.70 | 74.54 | 118.85 |
| Sub-PCA | MAD | 26.55 | 28.00 | 23.72 | 62.37 | 54.31 | 54.74 | _0.18_ |
| USAD | MAD | 23.92 | 34.54 | 24.46 | 59.84 | 61.60 | 55.24 | 1.57 |
| TimesFM | MAD | 25.47 | 52.18 | 24.98 | 66.32 | 94.65 | 75.72 | 130.09 |
| Chronos | MAD | 23.58 | _55.65_ | 25.08 | 64.87 | **96.33** | _75.90_ | 69.10 |
| IForest | MAD | _33.99_ | 26.62 | 26.79 | _68.24_ | 68.44 | 64.72 | 0.52 |
| SR | MAD | 32.61 | 40.77 | _30.57_ | 66.80 | _95.11_ | **75.94** | **0.02** |
| Our (GPT-4o) | – | **47.60** | 31.88 | **34.20** | **76.10** | 81.49 | 74.64 | 140.63 |

## E  Full benchmark results

| Gemini-1.5-Flash | | | Standard | | | Affiliation | | |
|---|---|---|---|---|---|---|---|---|
| Datasets | Code | Prompt | Prec. | Recall | F1 | Prec. | Recall | F1 |
| Trend | A | 1shot-vision-cot | 41.07 | 40.88 | 39.55 | 50.91 | 53.18 | 51.82 |
| | B | 1shot-vision-calc | 63.33 | 63.63 | 62.75 | 67.09 | 67.43 | 67.25 |
| | C | 1shot-vision-dyscalc | 62.01 | 62.22 | 61.86 | 64.12 | 64.37 | 64.24 |
| | D | 1shot-vision | 57.59 | 57.73 | 56.75 | 62.41 | 62.81 | 62.59 |
| | E | 0shot-vision-cot | 57.93 | 57.94 | 57.93 | 57.99 | 58.00 | 57.99 |
| | F | 0shot-vision-calc | 59.70 | 59.44 | 59.48 | 60.15 | 60.20 | 60.18 |
| | G | 0shot-vision-dyscalc | 59.08 | 59.03 | 58.95 | 59.43 | 59.47 | 59.45 |
| | H | 0shot-vision | 59.77 | 59.57 | 59.60 | 60.17 | 60.22 | 60.19 |
| | I | 1shot-text-s0.3-cot | 5.64 | 7.96 | 6.32 | 19.15 | 24.30 | 21.23 |
| | J | 1shot-text-s0.3 | 2.03 | 2.15 | 1.88 | 11.72 | 17.19 | 13.70 |
| | K | 0shot-text-s0.3-tpd | 0.00 | 0.00 | 0.00 | 3.03 | 5.81 | 3.97 |
| | L | 0shot-text-s0.3-pap | 0.00 | 0.00 | 0.00 | 3.37 | 7.25 | 4.56 |
| | M | 0shot-text-s0.3-dyscalc | 0.00 | 0.00 | 0.00 | 3.90 | 7.85 | 5.20 |
| | N | 0shot-text-s0.3-csv | 0.00 | 0.00 | 0.00 | 3.15 | 7.16 | 4.31 |
| | O | 0shot-text-s0.3-cot-tpd | 3.25 | 3.25 | 3.25 | 7.83 | 10.62 | 8.88 |
| | P | 0shot-text-s0.3-cot-pap | 0.25 | 0.25 | 0.25 | 3.26 | 5.66 | 4.07 |
| | Q | 0shot-text-s0.3-cot-csv | 0.50 | 0.50 | 0.50 | 5.51 | 9.70 | 6.88 |
| | R | 0shot-text-s0.3-cot | 0.52 | 1.01 | 0.63 | 7.83 | 13.90 | 9.94 |
| | S | 0shot-text-s0.3-calc | 0.00 | 0.00 | 0.00 | 4.59 | 8.98 | 6.06 |
| | T | 0shot-text-s0.3 | 2.12 | 2.58 | 2.07 | 12.28 | 17.76 | 14.31 |
| | U | 0shot-text | 0.00 | 0.00 | 0.00 | 3.99 | 7.79 | 5.27 |
| | Our | 0shot-text | 65.38 | 61.87 | 62.52 | 67.47 | 70.49 | 68.28 |
| | Our | 0shot-text-vision (w/o value) | 70.87 | 71.07 | 70.30 | _88.41_ | _91.37_ | _89.62_ |
| | Our | 0shot-text-vision (w/o index) | _74.95_ | _75.70_ | _74.38_ | 85.63 | 86.40 | 85.95 |
| | Our | 0shot-text-vision (w/o deseason) | 58.76 | 58.45 | 58.47 | 59.69 | 61.47 | 60.27 |
| | Our | 0shot-text-vision | **82.90** | **81.81** | **81.28** | **90.08** | **92.70** | **91.11** |
| | A | 1shot-vision-cot | 14.29 | 16.89 | 13.80 | 54.19 | 41.68 | 43.93 |
| | B | 1shot-vision-calc | 23.10 | 20.50 | 20.73 | 34.76 | 24.87 | 27.46 |
| | C | 1shot-vision-dyscalc | 23.73 | 21.83 | 21.61 | 36.35 | 24.37 | 27.57 |

| | | | | | | | | |
|---|---|---|---|---|---|---|---|---|
| | D | 1shot-vision | 21.78 | 21.31 | 20.52 | 35.84 | 25.89 | 28.46 |
| | E | 0shot-vision-cot | 12.65 | 12.71 | 12.45 | 14.16 | 12.52 | 12.98 |
| | F | 0shot-vision-calc | 17.82 | 19.68 | 18.14 | 23.65 | 17.80 | 19.39 |
| | G | 0shot-vision-dyscalc | 16.71 | 17.84 | 16.83 | 21.44 | 16.58 | 17.88 |
| | H | 0shot-vision | 13.56 | 13.77 | 13.54 | 15.47 | 13.15 | 13.76 |
| | I | 1shot-text-s0.3-cot | 12.81 | 17.30 | 12.31 | 56.67 | 50.63 | 51.30 |
| | J | 1shot-text-s0.3 | 10.20 | 14.73 | 10.45 | 56.37 | 52.78 | 52.61 |
| | K | 0shot-text-s0.3-tpd | 2.57 | 1.57 | 1.63 | 39.54 | 29.61 | 32.22 |
| | L | 0shot-text-s0.3-pap | 1.97 | 0.73 | 0.94 | 33.36 | 22.71 | 25.54 |
| | M | 0shot-text-s0.3-dyscalc | 3.74 | 3.24 | 2.91 | 36.95 | 28.19 | 30.32 |
| | N | 0shot-text-s0.3-csv | 2.77 | 2.21 | 2.08 | 46.48 | 33.66 | 37.03 |
| | O | 0shot-text-s0.3-cot-tpd | 2.61 | 2.37 | 1.97 | 25.96 | 19.14 | 21.00 |
| | P | 0shot-text-s0.3-cot-pap | 3.92 | 2.35 | 2.61 | 29.46 | 19.21 | 22.12 |
| | Q | 0shot-text-s0.3-cot-csv | 2.55 | 2.34 | 1.94 | 30.81 | 23.37 | 25.37 |
| | R | 0shot-text-s0.3-cot | 3.46 | 4.78 | 3.36 | 21.76 | 19.93 | 19.99 |
| | S | 0shot-text-s0.3-calc | 3.29 | 2.61 | 2.51 | 36.76 | 27.58 | 29.91 |
| | T | 0shot-text-s0.3 | 11.28 | 14.24 | 10.91 | 58.42 | 52.67 | 53.59 |
| | U | 0shot-text | 4.01 | 4.30 | 3.63 | 39.24 | 31.84 | 33.65 |
| | Our | 0shot-text | 37.92 | 24.04 | 25.75 | 62.71 | 52.37 | 54.95 |
| | Our | 0shot-text-vision (w/o value) | 30.17 | **50.64** | 33.71 | 80.53 | **70.80** | **73.25** |
| | Our | 0shot-text-vision (w/o index) | 30.20 | 45.50 | 32.56 | 77.49 | 67.14 | 69.69 |
| | Our | 0shot-text-vision (w/o deseason) | **48.87** | 33.96 | 36.87 | 55.99 | 41.00 | 45.47 |
| | Our | 0shot-text-vision | 46.23 | 47.13 | **39.98** | **81.57** | 70.16 | 73.03 |
| | A | 1shot-vision-cot | 43.09 | 67.37 | 49.31 | 82.62 | 83.34 | 82.61 |
| | B | 1shot-vision-calc | 55.95 | 77.36 | 61.54 | 94.88 | 94.55 | 94.17 |
| | C | 1shot-vision-dyscalc | 54.29 | 78.90 | 60.44 | 95.18 | 96.15 | 95.25 |
| | D | 1shot-vision | 53.54 | 78.76 | 59.96 | 93.42 | 93.83 | 93.19 |
| | E | 0shot-vision-cot | 51.87 | 84.03 | 59.20 | 91.78 | 89.43 | 89.75 |
| | F | 0shot-vision-calc | 51.56 | 87.56 | 59.74 | 94.28 | 95.88 | 94.63 |
| | G | 0shot-vision-dyscalc | 51.40 | 88.72 | 59.53 | 94.18 | 95.87 | 94.52 |
| | H | 0shot-vision | 52.35 | **90.69** | 60.90 | 94.22 | 95.46 | 94.38 |
| | I | 1shot-text-s0.3-cot | 8.19 | 9.11 | 8.05 | 45.60 | 43.79 | 43.26 |
| | J | 1shot-text-s0.3 | 3.63 | 4.20 | 3.61 | 43.49 | 41.13 | 40.93 |
| | K | 0shot-text-s0.3-tpd | 1.83 | 1.76 | 1.77 | 22.66 | 20.71 | 20.94 |
| | L | 0shot-text-s0.3-pap | 0.00 | 0.00 | 0.00 | 15.58 | 14.10 | 14.28 |
| Point | M | 0shot-text-s0.3-dyscalc | 0.20 | 0.10 | 0.12 | 18.22 | 18.78 | 18.08 |
| | N | 0shot-text-s0.3-csv | 3.81 | 2.85 | 2.74 | 34.27 | 29.22 | 30.44 |
| | O | 0shot-text-s0.3-cot-tpd | 3.89 | 3.79 | 3.76 | 18.41 | 16.85 | 17.14 |
| | P | 0shot-text-s0.3-cot-pap | 3.31 | 3.51 | 3.30 | 17.68 | 16.21 | 16.44 |
| | Q | 0shot-text-s0.3-cot-csv | 2.88 | 2.86 | 2.19 | 24.03 | 20.64 | 21.46 |
| | R | 0shot-text-s0.3-cot | 2.66 | 3.72 | 2.54 | 19.37 | 20.22 | 19.12 |
| | S | 0shot-text-s0.3-calc | 0.10 | 0.05 | 0.06 | 18.67 | 19.37 | 18.53 |
| | T | 0shot-text-s0.3 | 2.75 | 3.60 | 2.83 | 43.01 | 41.02 | 40.64 |
| | U | 0shot-text | 1.09 | 1.06 | 0.93 | 23.84 | 24.38 | 23.35 |
| | Our | 0shot-text | 67.99 | 35.56 | 43.39 | 69.75 | 67.17 | 67.79 |
| | Our | 0shot-text-vision (w/o value) | 56.52 | 81.42 | 62.76 | 95.86 | 95.95 | 95.52 |
| | Our | 0shot-text-vision (w/o index) | 57.21 | 76.78 | 62.52 | 93.78 | 96.26 | 94.54 |
| | Our | 0shot-text-vision (w/o deseason) | 87.85 | 88.36 | **85.44** | **98.91** | **97.12** | **97.64** |
| | Our | 0shot-text-vision | **90.89** | 74.93 | 78.41 | 98.45 | 96.81 | 97.11 |
| | A | 1shot-vision-cot | 22.85 | 50.51 | 28.87 | 69.16 | 71.62 | 70.05 |
| | B | 1shot-vision-calc | 36.28 | 64.25 | 42.40 | 81.41 | 84.05 | 82.40 |
| | C | 1shot-vision-dyscalc | 35.52 | 61.50 | 41.15 | 81.49 | 83.06 | 81.82 |
| | D | 1shot-vision | 33.77 | 63.17 | 40.02 | 80.05 | 82.67 | 81.07 |
| | E | 0shot-vision-cot | 28.58 | 61.32 | 34.67 | 75.91 | 73.62 | 73.75 |
| | F | 0shot-vision-calc | 20.96 | 60.42 | 28.02 | 68.04 | 70.94 | 69.07 |
| | G | 0shot-vision-dyscalc | 26.94 | 66.70 | 34.00 | 74.17 | 77.29 | 75.34 |
| | H | 0shot-vision | 32.70 | **74.46** | 40.19 | 79.35 | 83.16 | 80.95 |

Range

| Code | Prompt | Prec. | Recall | F1 | Prec. | Recall | F1 |
|---|---|---|---|---|---|---|---|
| I | 1shot-text-s0.3-cot | 12.04 | 14.57 | 12.31 | 46.38 | 49.43 | 46.57 |
| J | 1shot-text-s0.3 | 7.12 | 13.71 | 8.42 | 47.06 | 56.95 | 50.69 |
| K | 0shot-text-s0.3-tpd | 2.30 | 2.63 | 2.13 | 29.22 | 28.56 | 27.96 |
| L | 0shot-text-s0.3-pap | 1.97 | 0.88 | 1.08 | 30.66 | 27.80 | 28.09 |
| M | 0shot-text-s0.3-dyscalc | 2.20 | 1.95 | 1.85 | 33.94 | 36.51 | 34.22 |
| N | 0shot-text-s0.3-csv | 5.00 | 4.52 | 4.62 | 35.17 | 33.31 | 33.28 |
| O | 0shot-text-s0.3-cot-tpd | 2.87 | 3.03 | 2.86 | 16.75 | 14.98 | 15.26 |
| P | 0shot-text-s0.3-cot-pap | 6.59 | 6.17 | 6.17 | 25.46 | 22.12 | 23.01 |
| Q | 0shot-text-s0.3-cot-csv | 2.70 | 2.24 | 2.34 | 16.85 | 15.47 | 15.75 |
| R | 0shot-text-s0.3-cot | 4.28 | 4.90 | 4.18 | 20.54 | 22.27 | 20.84 |
| S | 0shot-text-s0.3-calc | 2.24 | 1.92 | 1.81 | 35.13 | 36.94 | 34.99 |
| T | 0shot-text-s0.3 | 7.15 | 12.76 | 8.34 | 46.77 | 56.31 | 50.25 |
| U | 0shot-text | 2.92 | 4.53 | 2.98 | 32.52 | 38.30 | 34.27 |
| Our | 0shot-text | **59.97** | 53.65 | 54.81 | 64.74 | 67.29 | 65.47 |
| Our | 0shot-text-vision (w/o value) | 38.54 | 67.64 | 44.81 | **83.56** | **86.21** | **84.57** |
| Our | 0shot-text-vision (w/o index) | 20.27 | 49.77 | 26.97 | 61.38 | 68.83 | 64.52 |
| Our | 0shot-text-vision (w/o deseason) | 54.54 | 61.60 | **55.60** | 66.94 | 68.64 | 67.60 |
| Our | 0shot-text-vision | 53.15 | 59.37 | 53.57 | 66.36 | 68.09 | 66.96 |

| InternVL2-Llama3-76B | | | Standard | | | Affiliation | |
|---|---|---|---|---|---|---|---|
| Datasets | Code | Prompt | Prec. | Recall | F1 | Prec. | Recall | F1 |
|---|---|---|---|---|---|---|---|---|
| Trend | A | 1shot-vision-cot | 44.90 | 44.90 | 44.29 | 46.92 | 47.32 | 47.08 |
| | B | 1shot-vision-calc | 33.76 | 39.43 | 32.67 | 40.33 | 44.13 | 41.61 |
| | C | 1shot-vision-dyscalc | 39.61 | 43.23 | 38.34 | 48.48 | 51.02 | 49.37 |
| | D | 1shot-vision | 38.98 | 42.27 | 37.93 | 47.54 | 50.00 | 48.43 |
| | E | 0shot-vision-cot | 51.13 | 51.34 | 51.06 | 53.27 | 54.16 | 53.60 |
| | F | 0shot-vision-calc | 41.31 | 44.80 | 41.82 | 46.76 | 52.22 | 48.64 |
| | G | 0shot-vision-dyscalc | 36.48 | 45.76 | 36.67 | 45.49 | 54.89 | 48.67 |
| | H | 0shot-vision | 27.27 | 45.87 | 27.82 | 37.77 | 49.50 | 41.68 |
| | I | 1shot-text-s0.3-cot | 37.44 | 36.65 | 35.74 | 42.01 | 42.32 | 42.14 |
| | J | 1shot-text-s0.3 | 30.18 | 31.05 | 29.77 | 36.14 | 37.59 | 36.68 |
| | K | 0shot-text-s0.3-tpd | 2.06 | 4.73 | 2.30 | 5.75 | 9.78 | 7.09 |
| | L | 0shot-text-s0.3-pap | 46.08 | 46.75 | 46.14 | 47.07 | 48.13 | 47.43 |
| | M | 0shot-text-s0.3-dyscalc | 11.38 | 13.71 | 11.63 | 14.30 | 17.62 | 15.41 |
| | N | 0shot-text-s0.3-csv | 0.50 | 0.50 | 0.50 | 5.45 | 9.52 | 6.83 |
| | O | 0shot-text-s0.3-cot-tpd | 10.76 | 11.00 | 10.77 | 11.39 | 12.00 | 11.60 |
| | P | 0shot-text-s0.3-cot-pap | 20.40 | 20.84 | 20.42 | 21.31 | 22.32 | 21.65 |
| | Q | 0shot-text-s0.3-cot-csv | 4.52 | 4.75 | 4.54 | 8.09 | 9.73 | 8.66 |
| | R | 0shot-text-s0.3-cot | 13.26 | 13.50 | 13.27 | 15.42 | 17.52 | 16.12 |
| | S | 0shot-text-s0.3-calc | 11.20 | 12.58 | 11.34 | 12.87 | 14.77 | 13.50 |
| | T | 0shot-text-s0.3 | 33.58 | 32.66 | 31.95 | 38.68 | 39.79 | 39.07 |
| | U | 0shot-text | 19.21 | 23.60 | 19.56 | 22.80 | 26.84 | 24.16 |
| | Our | 0shot-text | 32.32 | 52.20 | 35.72 | 47.85 | 54.93 | 50.59 |
| | Our | 0shot-text-vision | **56.90** | **66.60** | **58.83** | **68.36** | **74.21** | **70.60** |
| Freq | A | 1shot-vision-cot | 2.76 | 8.41 | 3.58 | 35.98 | 42.04 | 36.75 |
| | B | 1shot-vision-calc | 4.55 | 13.45 | 5.19 | 35.04 | 38.20 | 33.78 |
| | C | 1shot-vision-dyscalc | 4.11 | 12.28 | 5.37 | 35.10 | 38.41 | 33.73 |
| | D | 1shot-vision | 4.31 | 15.22 | 4.92 | 38.93 | 44.67 | 38.43 |
| | E | 0shot-vision-cot | 4.97 | 9.25 | 5.37 | 30.18 | 27.70 | 27.27 |
| | F | 0shot-vision-calc | 4.68 | 15.55 | 5.41 | 26.42 | 29.19 | 25.80 |
| | G | 0shot-vision-dyscalc | 7.16 | 18.74 | 8.88 | 32.78 | 33.72 | 31.01 |
| | H | 0shot-vision | 6.51 | 17.82 | 8.81 | 35.94 | 35.14 | 33.37 |
| | I | 1shot-text-s0.3-cot | 3.05 | 2.69 | 2.40 | 30.54 | 26.81 | 26.86 |
| | J | 1shot-text-s0.3 | 4.24 | 4.93 | 3.22 | 37.50 | 32.79 | 32.66 |

| | | | | | | | | |
|---|---|---|---|---|---|---|---|---|
| | K | 0shot-text-s0.3-tpd | 4.01 | 7.60 | 4.01 | 24.88 | 21.46 | 21.45 |
| | L | 0shot-text-s0.3-pap | 8.36 | 9.29 | 8.38 | 13.77 | 12.68 | 12.77 |
| | M | 0shot-text-s0.3-dyscalc | 7.15 | 8.20 | 7.22 | 16.55 | 13.84 | 14.39 |
| | N | 0shot-text-s0.3-csv | 3.65 | 8.97 | 3.62 | 37.52 | 31.89 | 32.68 |
| | O | 0shot-text-s0.3-cot-tpd | 3.13 | 3.73 | 3.15 | 10.65 | 8.46 | 8.98 |
| | P | 0shot-text-s0.3-cot-pap | 3.79 | 3.68 | 3.71 | 6.45 | 5.57 | 5.78 |
| | Q | 0shot-text-s0.3-cot-csv | 2.60 | 4.52 | 2.69 | 19.44 | 17.81 | 17.71 |
| | R | 0shot-text-s0.3-cot | 6.51 | 7.71 | 6.72 | 12.89 | 11.98 | 12.06 |
| | S | 0shot-text-s0.3-calc | 8.18 | 9.48 | 8.30 | 14.36 | 13.50 | 13.45 |
| | T | 0shot-text-s0.3 | 3.38 | 7.18 | 3.33 | 37.12 | 33.84 | 33.16 |
| | U | 0shot-text | 5.66 | 12.39 | 6.04 | 15.82 | 17.30 | 15.53 |
| | Our | 0shot-text | **20.35** | **48.90** | **19.72** | **60.36** | 63.00 | 57.04 |
| | Our | 0shot-text-vision | 13.42 | 46.70 | 17.22 | 57.58 | 73.33 | **62.47** |
| | A | 1shot-vision-cot | 3.87 | 12.65 | 4.61 | 33.66 | 45.65 | 37.30 |
| | B | 1shot-vision-calc | 10.01 | 29.92 | 12.26 | 44.34 | 57.93 | 48.86 |
| | C | 1shot-vision-dyscalc | 8.98 | 31.81 | 11.90 | 42.77 | 59.47 | 48.65 |
| | D | 1shot-vision | 9.97 | 29.84 | 12.99 | 44.76 | 57.62 | 49.26 |
| | E | 0shot-vision-cot | 4.35 | 12.95 | 5.47 | 31.51 | 39.00 | 33.61 |
| | F | 0shot-vision-calc | 22.97 | 58.75 | 28.77 | **66.27** | **76.14** | **70.44** |
| | G | 0shot-vision-dyscalc | 17.31 | 54.36 | 23.36 | 59.49 | 70.15 | 63.79 |
| | H | 0shot-vision | 14.26 | 53.10 | 20.64 | 56.18 | 67.83 | 60.77 |
| | I | 1shot-text-s0.3-cot | 10.20 | 9.98 | 9.43 | 30.63 | 30.69 | 29.49 |
| | J | 1shot-text-s0.3 | 8.69 | 9.89 | 7.70 | 32.37 | 32.95 | 31.19 |
| | K | 0shot-text-s0.3-tpd | 7.00 | 7.00 | 7.00 | 14.89 | 15.08 | 14.88 |
| Point | L | 0shot-text-s0.3-pap | 25.26 | 25.50 | 25.27 | 27.14 | 27.26 | 27.16 |
| | M | 0shot-text-s0.3-dyscalc | 18.89 | 19.66 | 18.94 | 23.35 | 23.80 | 23.46 |
| | N | 0shot-text-s0.3-csv | 11.92 | 13.91 | 11.20 | 40.05 | 38.01 | 37.78 |
| | O | 0shot-text-s0.3-cot-tpd | 9.21 | 9.82 | 8.81 | 26.12 | 24.75 | 24.70 |
| | P | 0shot-text-s0.3-cot-pap | 10.58 | 10.83 | 10.58 | 11.56 | 11.76 | 11.62 |
| | Q | 0shot-text-s0.3-cot-csv | 9.00 | 9.00 | 9.00 | 11.69 | 11.68 | 11.63 |
| | R | 0shot-text-s0.3-cot | 15.82 | 16.50 | 15.86 | 18.51 | 19.07 | 18.68 |
| | S | 0shot-text-s0.3-calc | 21.42 | 23.25 | 21.52 | 25.62 | 26.72 | 25.93 |
| | T | 0shot-text-s0.3 | 10.21 | 11.34 | 9.96 | 33.19 | 34.47 | 32.60 |
| | U | 0shot-text | 19.26 | 22.48 | 19.47 | 25.08 | 27.32 | 25.84 |
| | Our | 0shot-text | **37.76** | 53.21 | **37.08** | 59.81 | 65.15 | 60.64 |
| | Our | 0shot-text-vision | 28.86 | **60.61** | 34.92 | 56.23 | 69.93 | 61.22 |
| | A | 1shot-vision-cot | 3.57 | 5.90 | 3.66 | 31.10 | 34.97 | 31.58 |
| | B | 1shot-vision-calc | 19.32 | 35.23 | 21.91 | 60.17 | 65.54 | 61.65 |
| | C | 1shot-vision-dyscalc | 17.24 | 34.25 | 20.14 | 59.85 | 65.48 | 61.53 |
| | D | 1shot-vision | 17.79 | 33.39 | 20.35 | 58.16 | 61.04 | 58.53 |
| | E | 0shot-vision-cot | 8.25 | 12.05 | 8.64 | 34.21 | 36.28 | 33.99 |
| | F | 0shot-vision-calc | **27.58** | 56.64 | **32.82** | 72.83 | 78.02 | 74.93 |
| | G | 0shot-vision-dyscalc | 4.39 | 13.78 | 5.03 | 32.84 | 39.45 | 34.03 |
| | H | 0shot-vision | 27.08 | 56.67 | 32.58 | **73.52** | **79.03** | **75.84** |
| | I | 1shot-text-s0.3-cot | 8.70 | 8.90 | 8.48 | 28.66 | 28.08 | 27.47 |
| | J | 1shot-text-s0.3 | 8.61 | 9.95 | 8.37 | 27.64 | 26.60 | 25.94 |
| | K | 0shot-text-s0.3-tpd | 7.83 | 12.72 | 8.05 | 22.23 | 23.44 | 21.88 |
| Range | L | 0shot-text-s0.3-pap | 25.43 | 27.34 | 25.51 | 28.87 | 29.71 | 29.00 |
| | M | 0shot-text-s0.3-dyscalc | 7.56 | 16.99 | 7.92 | 21.17 | 25.37 | 22.00 |
| | N | 0shot-text-s0.3-csv | 7.83 | 8.24 | 7.49 | 38.79 | 34.59 | 35.52 |
| | O | 0shot-text-s0.3-cot-tpd | 6.40 | 6.46 | 6.34 | 11.74 | 11.09 | 11.12 |
| | P | 0shot-text-s0.3-cot-pap | 12.58 | 12.40 | 12.37 | 18.20 | 17.09 | 17.31 |
| | Q | 0shot-text-s0.3-cot-csv | 4.83 | 5.71 | 4.75 | 19.71 | 18.08 | 18.24 |
| | R | 0shot-text-s0.3-cot | 9.85 | 10.45 | 9.94 | 16.89 | 17.87 | 17.13 |
| | S | 0shot-text-s0.3-calc | 10.49 | 13.55 | 10.29 | 20.98 | 21.34 | 20.46 |
| | T | 0shot-text-s0.3 | 7.45 | 8.89 | 7.48 | 25.94 | 25.18 | 24.59 |
| | U | 0shot-text | 8.95 | 27.30 | 9.39 | 27.16 | 36.32 | 29.81 |

| | | | Standard | | | Affiliation | | |
|---|---|---|---|---|---|---|---|---|
| Our | 0shot-text | | 20.27 | 54.87 | 24.33 | 49.22 | 59.77 | 51.89 |
| Our | 0shot-text-vision | | 20.82 | **63.50** | 27.68 | 50.37 | 69.70 | 57.33 |

| GPT-4o | | | Standard | | | Affiliation | | |
|---|---|---|---|---|---|---|---|---|
| Datasets | Code | Prompt | Prec. | Recall | F1 | Prec. | Recall | F1 |
| Trend | A | 1shot-vision-cot | 57.50 | 57.50 | 57.50 | 57.50 | 57.50 | 57.50 |
| | H | 0shot-vision | 57.50 | 57.50 | 57.50 | 57.50 | 57.50 | 57.50 |
| | U | 0shot-text | 6.67 | 5.91 | 6.04 | 14.85 | 22.53 | 17.48 |
| | Our | 0shot-text | **83.47** | 68.97 | 70.84 | **87.48** | **87.93** | **87.41** |
| | Our | 0shot-text-vision (w/o value) | 58.13 | 57.82 | 57.90 | 58.23 | 58.21 | 58.22 |
| | Our | 0shot-text-vision (w/o index) | 64.25 | 64.40 | 63.80 | 71.40 | 71.87 | 71.63 |
| | Our | 0shot-text-vision (w/o deseason) | 57.50 | 57.50 | 57.50 | 57.50 | 57.50 | 57.50 |
| | Our | 0shot-text-vision | 79.12 | **75.48** | **76.27** | 79.66 | 79.43 | 79.54 |
| Freq | A | 1shot-vision-cot | 15.66 | 17.93 | 16.12 | 28.10 | 22.09 | 23.65 |
| | H | 0shot-vision | 13.68 | 15.69 | 14.27 | 19.02 | 16.13 | 16.98 |
| | U | 0shot-text | 15.01 | 10.54 | 11.20 | 34.05 | 24.71 | 27.40 |
| | Our | 0shot-text | **50.38** | 27.15 | 29.34 | **72.56** | **60.06** | **62.88** |
| | Our | 0shot-text-vision (w/o value) | 16.27 | 20.95 | 16.82 | 37.87 | 31.69 | 33.40 |
| | Our | 0shot-text-vision (w/o index) | 16.80 | 20.87 | 17.06 | 52.02 | 41.44 | 44.39 |
| | Our | 0shot-text-vision (w/o deseason) | 37.86 | 19.01 | 22.13 | 39.54 | 27.41 | 30.80 |
| | Our | 0shot-text-vision | 45.42 | **36.09** | **35.16** | 65.85 | 51.44 | 55.66 |
| Point | A | 1shot-vision-cot | 30.12 | 46.21 | 33.57 | 72.69 | 73.24 | 72.08 |
| | H | 0shot-vision | 45.20 | 68.00 | 50.48 | 87.71 | 87.70 | 87.09 |
| | U | 0shot-text | 33.49 | 32.11 | 32.31 | 73.21 | 73.64 | 72.87 |
| | Our | 0shot-text | 72.50 | 36.66 | 44.00 | 74.48 | 74.86 | 73.52 |
| | Our | 0shot-text-vision (w/o value) | 46.67 | 63.10 | 50.96 | 87.29 | 85.85 | 85.87 |
| | Our | 0shot-text-vision (w/o index) | 44.10 | 44.11 | 43.30 | 86.06 | 82.98 | 83.65 |
| | Our | 0shot-text-vision (w/o deseason) | **98.87** | **90.33** | **93.57** | **99.96** | **97.76** | **98.54** |
| | Our | 0shot-text-vision | 90.51 | 73.14 | 78.71 | 92.66 | 91.48 | 91.79 |
| Range | A | 1shot-vision-cot | 22.65 | 32.35 | 24.52 | 67.43 | 70.52 | 68.34 |
| | H | 0shot-vision | 41.76 | 61.19 | 45.87 | 84.52 | 86.62 | 85.15 |
| | U | 0shot-text | 23.58 | 21.90 | 21.47 | 63.27 | 61.17 | 61.18 |
| | Our | 0shot-text | 72.56 | 66.75 | 67.40 | 76.20 | 77.96 | 76.43 |
| | Our | 0shot-text-vision (w/o value) | 42.21 | 58.69 | 45.99 | 83.33 | 83.35 | 82.81 |
| | Our | 0shot-text-vision (w/o index) | 41.77 | 46.07 | 42.70 | 84.21 | 83.73 | 83.44 |
| | Our | 0shot-text-vision (w/o deseason) | **96.39** | 88.93 | **91.67** | **97.75** | **96.20** | **96.72** |
| | Our | 0shot-text-vision | 92.70 | **88.95** | 90.00 | 94.25 | 93.05 | 93.45 |

| Qwen2.5-VL-72B-Instruct | | | Standard | | | Affiliation | | |
|---|---|---|---|---|---|---|---|---|
| Datasets | Code | Prompt | Prec. | Recall | F1 | Prec. | Recall | F1 |
| Trend | A | 1shot-vision-cot | 54.59 | 55.00 | 54.66 | 54.94 | 55.00 | 54.97 |
| | B | 1shot-vision-calc | 43.78 | 44.79 | 42.79 | 48.13 | 48.69 | 48.37 |
| | C | 1shot-vision-dyscalc | 42.98 | 46.53 | 42.82 | 49.49 | 50.76 | 50.03 |
| | E | 0shot-vision-cot | 55.93 | 55.89 | 55.81 | 56.19 | 56.22 | 56.20 |
| | F | 0shot-vision-calc | 61.30 | 60.87 | 60.69 | 62.07 | 62.06 | 62.06 |
| | G | 0shot-vision-dyscalc | 60.75 | 60.74 | 60.43 | 61.56 | 61.62 | 61.58 |
| | H | 0shot-vision | 60.19 | 59.95 | 59.79 | 60.64 | 60.64 | 60.64 |
| | J | 1shot-text-s0.3 | 45.52 | 45.51 | 45.52 | 47.48 | 47.99 | 47.69 |
| | L | 0shot-text-s0.3-pap | 40.50 | 40.50 | 40.50 | 40.76 | 40.84 | 40.79 |
| | M | 0shot-text-s0.3-dyscalc | 10.25 | 10.25 | 10.25 | 10.98 | 11.38 | 11.13 |

| | | | | | | | |
|---|---|---|---|---|---|---|---|
| | N | 0shot-text-s0.3-csv | 16.75 | 16.75 | 16.75 | 19.08 | 20.10 | 19.44 |
| | P | 0shot-text-s0.3-cot-pap | 18.75 | 18.75 | 18.75 | 19.01 | 19.15 | 19.06 |
| | S | 0shot-text-s0.3-calc | 8.81 | 8.80 | 8.81 | 10.03 | 10.88 | 10.33 |
| | T | 0shot-text-s0.3 | 46.81 | 46.77 | 46.78 | 48.76 | 49.48 | 49.02 |
| | U | 0shot-text | 56.00 | 56.00 | 56.00 | 56.08 | 56.14 | 56.11 |
| | Our | 0shot-text | **86.06** | **83.24** | **82.00** | **91.96** | **93.96** | **92.69** |
| | Our | 0shot-text-vision (w/o value) | 61.38 | 60.66 | 60.77 | 62.70 | 62.86 | 62.77 |
| | Our | 0shot-text-vision (w/o index) | 66.82 | 68.73 | 66.34 | 81.77 | 84.62 | 82.88 |
| | Our | 0shot-text-vision (w/o deseason) | 60.80 | 59.36 | 59.68 | 61.37 | 61.27 | 61.31 |
| | Our | 0shot-text-vision | 85.61 | 81.49 | 81.09 | 89.95 | 91.19 | 90.20 |
| Freq | A | 1shot-vision-cot | 5.16 | 7.29 | 5.35 | 16.12 | 14.46 | 14.67 |
| | B | 1shot-vision-calc | 10.98 | 11.84 | 10.00 | 28.72 | 23.05 | 23.95 |
| | C | 1shot-vision-dyscalc | 10.15 | 11.80 | 9.17 | 31.71 | 26.34 | 26.81 |
| | E | 0shot-vision-cot | 9.33 | 13.26 | 9.93 | 17.26 | 16.61 | 16.33 |
| | F | 0shot-vision-calc | 14.29 | 15.52 | 14.15 | 31.78 | 24.87 | 26.59 |
| | G | 0shot-vision-dyscalc | 16.40 | 18.10 | 16.39 | 31.99 | 24.42 | 26.38 |
| | H | 0shot-vision | 15.47 | 18.62 | 15.75 | 29.05 | 26.55 | 26.64 |
| | J | 1shot-text-s0.3 | 16.39 | 16.31 | 15.45 | 63.17 | 50.72 | 54.66 |
| | L | 0shot-text-s0.3-pap | 6.75 | 6.75 | 6.75 | 7.98 | 7.36 | 7.52 |
| | M | 0shot-text-s0.3-dyscalc | 3.72 | 3.56 | 3.50 | 19.20 | 14.15 | 15.51 |
| | N | 0shot-text-s0.3-csv | 11.16 | 9.81 | 9.98 | 33.34 | 23.73 | 26.37 |
| | P | 0shot-text-s0.3-cot-pap | 4.81 | 4.82 | 4.81 | 5.26 | 5.06 | 5.12 |
| | S | 0shot-text-s0.3-calc | 5.51 | 4.96 | 4.76 | 19.25 | 13.76 | 15.27 |
| | T | 0shot-text-s0.3 | 19.30 | 18.79 | 18.14 | 64.73 | 52.41 | 56.38 |
| | U | 0shot-text | 11.45 | 10.11 | 10.20 | 18.80 | 14.19 | 15.43 |
| | Our | 0shot-text | **56.91** | 30.77 | 33.57 | **80.90** | 62.84 | 67.54 |
| | Our | 0shot-text-vision (w/o value) | 23.87 | 28.34 | 22.97 | 61.32 | 51.76 | 53.90 |
| | Our | 0shot-text-vision (w/o index) | 21.97 | 27.84 | 21.22 | 63.38 | 53.50 | 55.76 |
| | Our | 0shot-text-vision (w/o deseason) | 42.81 | 28.58 | 31.05 | 47.43 | 37.21 | 40.30 |
| | Our | 0shot-text-vision | 49.41 | **41.14** | **37.88** | 79.18 | **64.95** | **68.57** |
| Point | A | 1shot-vision-cot | 18.47 | 22.57 | 18.98 | 35.89 | 38.85 | 36.88 |
| | B | 1shot-vision-calc | 41.02 | 43.32 | 40.04 | 73.97 | 79.19 | 75.82 |
| | C | 1shot-vision-dyscalc | 37.43 | 42.19 | 37.49 | 70.08 | 75.13 | 71.84 |
| | E | 0shot-vision-cot | 25.04 | 32.84 | 26.86 | 44.71 | 46.66 | 45.33 |
| | F | 0shot-vision-calc | 46.25 | 51.14 | 46.36 | 82.45 | 83.61 | 82.77 |
| | G | 0shot-vision-dyscalc | 47.30 | 53.39 | 47.98 | 82.86 | 84.52 | 83.41 |
| | H | 0shot-vision | 60.09 | 67.56 | 61.62 | 94.93 | 96.19 | 95.34 |
| | J | 1shot-text-s0.3 | 33.19 | 31.96 | 32.18 | 70.90 | 65.03 | 66.95 |
| | L | 0shot-text-s0.3-pap | 16.25 | 16.25 | 16.25 | 18.21 | 17.68 | 17.83 |
| | M | 0shot-text-s0.3-dyscalc | 12.62 | 12.19 | 12.27 | 26.96 | 23.93 | 24.80 |
| | N | 0shot-text-s0.3-csv | 18.06 | 16.77 | 17.03 | 39.32 | 34.06 | 35.67 |
| | P | 0shot-text-s0.3-cot-pap | 8.50 | 8.50 | 8.50 | 8.85 | 8.70 | 8.75 |
| | S | 0shot-text-s0.3-calc | 6.61 | 6.36 | 6.41 | 18.27 | 15.88 | 16.57 |
| | T | 0shot-text-s0.3 | 29.66 | 29.23 | 29.16 | 68.38 | 62.84 | 64.63 |
| | U | 0shot-text | 30.33 | 29.85 | 29.83 | 62.56 | 54.68 | 57.09 |
| | Our | 0shot-text | **94.52** | 56.07 | 63.16 | 98.31 | 91.50 | 93.78 |
| | Our | 0shot-text-vision (w/o value) | 56.83 | 58.55 | 55.57 | 92.67 | 90.89 | 91.23 |
| | Our | 0shot-text-vision (w/o index) | 52.61 | 54.97 | 52.19 | 89.04 | 89.42 | 88.50 |
| | Our | 0shot-text-vision (w/o deseason) | 90.61 | **95.63** | **91.74** | **99.36** | **99.53** | **99.36** |
| | Our | 0shot-text-vision | 94.19 | 83.80 | 85.98 | 99.01 | 96.94 | 97.70 |
| Range | A | 1shot-vision-cot | 30.14 | 31.44 | 29.92 | 51.12 | 51.95 | 51.00 |
| | B | 1shot-vision-calc | 49.75 | 51.22 | 48.58 | **94.28** | 91.52 | 92.25 |
| | C | 1shot-vision-dyscalc | 49.84 | 51.61 | 48.97 | 94.03 | 90.73 | 91.68 |
| | E | 0shot-vision-cot | 36.46 | 43.52 | 37.61 | 61.41 | 61.55 | 61.27 |
| | F | 0shot-vision-calc | 46.03 | 55.76 | 48.02 | 92.50 | 91.52 | 91.57 |
| | G | 0shot-vision-dyscalc | 46.97 | 58.48 | 49.17 | 91.06 | 91.02 | 90.67 |

| | | | | | | | |
|---|---|---|---|---|---|---|---|
| H | 0shot-vision | 47.35 | 57.86 | 49.43 | 93.36 | **92.56** | **92.49** |
| J | 1shot-text-s0.3 | 14.26 | 16.71 | 14.76 | 50.94 | 55.36 | 52.01 |
| L | 0shot-text-s0.3-pap | 9.65 | 9.30 | 9.39 | 22.67 | 20.76 | 21.09 |
| M | 0shot-text-s0.3-dyscalc | 2.35 | 2.30 | 2.31 | 8.47 | 7.66 | 7.84 |
| N | 0shot-text-s0.3-csv | 9.77 | 9.14 | 9.20 | 32.94 | 28.68 | 29.93 |
| P | 0shot-text-s0.3-cot-pap | 3.94 | 3.33 | 3.40 | 6.23 | 5.32 | 5.52 |
| S | 0shot-text-s0.3-calc | 0.69 | 0.56 | 0.59 | 3.37 | 2.92 | 3.03 |
| T | 0shot-text-s0.3 | 11.42 | 13.58 | 11.85 | 46.77 | 51.10 | 47.91 |
| U | 0shot-text | 6.19 | 6.68 | 5.93 | 45.32 | 44.74 | 43.80 |
| Our | 0shot-text | 52.40 | 46.27 | 45.78 | 60.49 | 65.31 | 61.84 |
| Our | 0shot-text-vision (w/o value) | 47.70 | 53.44 | 48.04 | 91.94 | 89.46 | 90.06 |
| Our | 0shot-text-vision (w/o index) | 13.26 | 22.41 | 15.17 | 53.77 | 61.26 | 56.37 |
| Our | 0shot-text-vision (w/o deseason) | **61.49** | 66.94 | **62.43** | 70.08 | 73.02 | 71.26 |
| Our | 0shot-text-vision | 59.79 | **67.18** | 61.21 | 69.91 | 73.02 | 71.03 |

