# OpenReview forum: "Delving into Large Language Models for Effective Time-Series Anomaly Detection"
_NeurIPS.cc/2025/Conference — NeurIPS 2025 poster_

### Official Review · Reviewer_7DTS · 2025-06-29

**Clarity:** 3
**Significance:** 2
**Originality:** 2
**Rating:** 5
**Confidence:** 3

**Summary:**

This paper focuses on the problem of time series anomaly detection using (zero-shot) large language models. Similar to AnomLLM [1], it primarily considers four types of anomalies: Trend Shift, Frequency Change, Contextual Point, and Out-of-Range. The authors observe that large language models struggle not only with determining whether a sequence is anomalous, but also—with even greater difficulty—with localizing the anomaly once it is detected. To improve performance, the proposed approach consists of three main strategies: 1. De-seasonalizing the time series, 2. Incorporating the visual modality of the sequence as additional input, and 3. Explicitly encoding the positional index of each value within the prompt. Experiments on several datasets report performance comparable to other strong baselines.

**Questions:**

1.	See the "Weaknesses" section.
2.	In the discussion on anomaly localization, the authors annotate the anomaly positions directly in the images. I wonder whether this might influence the judgment of the multimodal model, especially given the question-answering context.
3.	Figure 2 suggests that detecting trend anomalies is particularly challenging, yet Figure 3 (b) shows that their localization performance is actually the best. What accounts for this discrepancy?
4.	As discussed in [4], a simplified model that removes the large language model component and uses only a few lightweight network layers can still achieve strong performance. Could the authors conduct similar anomaly detection experiments to evaluate this alternative?

**References:**

[1] Zhou, Zihao, and Rose Yu. "Can LLMs Understand Time Series Anomalies?."
[2] Cao, Defu, et al. "Tempo: Prompt-based generative pre-trained transformer for time series forecasting."
[3] Gruver, Nate, et al. "Large language models are zero-shot time series forecasters."
[4] Tan, Mingtian, et al. "Are language models actually useful for time series forecasting?."

**Ethical Concerns:**

["NO or VERY MINOR ethics concerns only"]

**Final Justification:**

After thorough evaluation of the rebuttal and discussions with authors, other reviewers, and the Area Chair (AC), I am pleased to recommend acceptance of this paper. The authors have effectively addressed the majority of the concerns raised during the initial review. Specifically, the authors explained to me the reasons for the inconsistencies in certain figures. Considering the current hot topic of large language model for anomaly detection and the paper's solid empirical study, I am personally inclined to recommend that this paper should be accepted.

**Paper Formatting Concerns:**

See the "Weaknesses" section.

**Quality:**

2

**Strengths And Weaknesses:**

Strength:
1.	The problem of LLMs to time series anomaly detection is a timely and rapidly growing research frontier.
2.	The paper is generally well-written and easy to follow, with clear organization and coherent argumentation.
3.	The paper presents well-crafted visualizations and conducts a thorough set of experiments. The study explores a diverse set of prompting strategies and representation techniques. It also uncovers several interesting and noteworthy findings.

Weaknesses:
1.	The analysis presented on “Why do LLMs fail at TSAD?” appears to closely align with issues already examined in prior work [1], offering limited new insight. Additionally, using anomaly detection (whether it exists or not) as a proxy to evaluate the understanding capabilities of large language models is somewhat unconvincing. However, the focus on the localization challenge represents a more compelling and well-motivated contribution.
2.	The main components of the proposed method have largely been explored in previous work—for example, the trend decomposition resembles that of Tempo [2], and the visual representation is similar to AnomLLM [1]. The index-aware prompting appears to be a novel design, though it bears some resemblance to the idea in [3]. Nevertheless, I have concerns about the potential overhead it may introduce in long-sequence scenarios.
3.	From the experimental section of the paper: “While the method performs reasonably well even in the absence of domain-specific training, its inference speed is significantly slower because of the overhead of decoding anomaly segments into textual descriptions.”

---

> ### Author Rebuttal · Authors · 2025-07-30
>
> > **W1) Concerns on the problem settings.**
>
> We sincerely thank the reviewer 7DTS for  insightful and constructive feedback. We would like to clarify the motivation behind our analysis, particularly regarding the instance-level anomaly detection task.
>
> 1. _Complementary to AnomLLM, with a focus on diagnosis:_
> AnomLLM provides an excellent foundation by benchmarking the capabilities and limitations of various LLMs across different anomaly types, validating common observations (for example, spike anomalies are easy while trend anomalies are hard). However, it does not explore why certain anomaly types are more challenging for LLMs, nor does it offer insight into how these limitations might be addressed. In contrast, our work shifts the focus from identifying what fails to understanding why it fails. We believe this makes a meaningful contribution to the community.
>
>
> 2. _Instance-level detection as a lower-bound diagnostic:_
> Our instance-level evaluation is not just about simplifying the task for LLMs, but about enabling a more fine-grained analysis of failure causes. Existing evaluation of TSAD often treats anomaly detection as a single end-to-end task, making it difficult to disentangle whether failures arise from a lack of temporal understanding or from imprecise localization. To address this, we explicitly isolate the LLM's understanding ability by designing an instance-level classification task (Figure 2 and Section 2.1) that simply asks whether any anomaly exists in a given sequence, without requiring precise localization. This removes the positional burden and serves as a lower-bound probe: if a model fails here, it likely lacks a basic grasp of the underlying time-series structure. While some of our findings are consistent with those of AnomLLM, our setup makes it easier to separate understanding from localization errors.
>
> > **W2) Concerns on the proposed method.**
>
> While we agree that the individual components such as trend decomposition and visual prompts have appeared in prior work, our paper does not aim to propose a novel method in terms of architectural complexity or algorithmic innovation.
> Instead, our main contribution lies in a systematic analysis of why LLMs underperform in TSAD, disentangling the issues of temporal understanding and localization.
> This analysis not only uncovers fundamental limitations in current LLM prompting practices, but also naturally leads to our proposed solution, which yields significant gains despite its simplicity.
>
> For the computational overhead, while adding indices increases the input context length, the main computational bottleneck typically lies in the decoding phase. As shown in our main results (Table 3), the overall runtime is not always significantly slower than the index-free prompt.
>
>
> > **W3) Concerns on computational limitation of LLM-based TSAD.**
>
> As noted in Section 5.1, current LLM based TSAD methods show clear limitations in inference efficiency, especially when compared to traditional lightweight models. However, given that LLMs were not originally designed for time series data, we believe it may be premature to interpret this limitation as a decisive weakness. If LLMs can demonstrate meaningful advantages in terms of accuracy or representational capacity for time series tasks, efficiency could naturally become a key target for future optimization. Rather than evaluating the approach solely based on its current limitations, we hope it can be viewed with an eye toward its potential for further development.
>
>
> > **Q2) In the discussion on anomaly localization, the authors annotate the anomaly positions directly in the images. I wonder whether this might influence the judgment of the multimodal model, especially given the question-answering context.**
>
> It’s true that the anomaly regions were visually annotated in the images, and this was an intentional design choice. The goal of this experiment (Section 2.2, Figure 3) was to remove the need for time-series understanding by providing ground-truth anomaly locations, thereby isolating the model’s localization ability. While trend anomalies showed near-perfect improvement under this setting, the model still failed significantly on point, range, and frequency anomalies. This highlights our key finding: even when LLMs are told where the anomalies are, they often struggle to translate that into precise start and end points, suggesting that localization itself remains a core challenge for LLM-based TSAD.
>
> >  **Q3)  Figure 2 suggests that detecting trend anomalies is particularly challenging, yet Figure 3 (b) shows that their localization performance is actually the best. What accounts for this discrepancy?**
>
> We would like to clarify the reviewer's observation regarding Figure 2 and Figure 3(b). While the results may appear inconsistent at first glance, they reflect differences in task setup. Figure 2 evaluates anomaly detection **without any guidance**, whereas Figure 3(b) focuses on anomaly detection **with ground truth provided**.
>
> Specifically, Figure 2 focuses on the instance-level anomaly detection task, where the model must determine whether any anomaly exists in the input sequence, without being told where or how many anomalies are present. This setup is used to reveal that there are still anomalies that LLMs fail to detect, even in problems that are simpler than TSAD, where the goal is to isolate the model’s ability to understand temporal patterns without the influence of localization.
>
> In contrast, Figure 3(b) evaluates the LLM's localization capability under a guided setting, where the anomaly regions are explicitly highlighted (e.g., in the input image). In this case, the model is relieved from the burden of understanding and only needs to translate the visual cues into concrete coordinate predictions. As a result, we observe substantial performance improvements, particularly for trend anomalies. This can be interpreted as the LLM previously struggling to recognize trend anomalies on its own, but in Figure 3, the anomaly locations were explicitly highlighted, allowing the model to rely on that information.
>
> In short, the results are not contradictory; instead, the ground truth provided in Figure 3 helps mitigate the challenge highlighted in Figure 2. We will update the manuscript to make this connection clearer and help readers better understand the intent behind the two experimental settings.
>
> > **Q4) As discussed in [4], a simplified model that removes the large language model component and uses only a few lightweight network layers can still achieve strong performance. Could the authors conduct similar anomaly detection experiments to evaluate this alternative?**
>
> We sincerely thank the reviewer for the thoughtful suggestion regarding the comparison to lightweight alternatives, such as those explored in [4]. Specifically, [4] focuses on forecasting tasks, where LLMs are alignment-tuned and tasked with generating future time series values. The conclusion that pretrained language model parameters offer limited benefit in that context is interesting, but it applies primarily to supervised and fine-tuned forecasting scenarios.
>
> In contrast, our work focuses on zero-shot anomaly detection, where no learning or task-specific tuning is involved. And we believe is a more suitable paradigm for real-world TSAD applications due to the scarcity of labeled anomalies. Our goal is not just to replicate the performance of conventional models, but to ultimately leverage the reasoning capability of LLMs, particularly their ability to interpret domain priors and to enable context-aware filtering, as preliminarily explored in Section 5.2. In this respect, simply replacing LLMs with lightweight transformers would overlook the fundamental motivation of our approach and sacrifice their essential reasoning abilities.
>
> However, we completely agree that comparing against simpler and more efficient methods is critical. In Section 5.1 and Figure 7, we already include a comparison against a wide range of baselines, many of which are significantly lighter than the transformer models explored in [4], including non-parametric methods (e.g., SR, IForest, MatrixProfile) and shallow neural models (e.g., CNN, AutoEncoder, USAD). While our method achieves stronger F1 performance, we also acknowledge and highlight in the paper that these lighter models remain more practical in latency-constrained environments.

---

> > ### Comment · Reviewer_7DTS · 2025-08-03
> >
> > Thank the authors for their clarification! I’m keeping my (positive) score — still, a good work!

---

> > > ### Author Response · Authors · 2025-08-08
> > >
> > > We truly appreciate your positive feedback and continued support. We will incorporate your suggestions to make the work even stronger.

---

### Official Review · Reviewer_K63Q · 2025-06-30

**Clarity:** 4
**Significance:** 3
**Originality:** 3
**Rating:** 5
**Confidence:** 2

**Summary:**

This paper investigates the limitations of Large Language Models in time series anomaly detection, pinpointing two key weaknesses: poor performance in identifying anomalous trend shifts and frequency changes, and inadequate anomaly localization. To address these issues, the authors propose a solution: plug-in de-seasonalization and index-aware prompting. Experimental results validate the effectiveness of these proposed methods.

**Questions:**

It would be good to highlight some failure cases of the proposed approach? Especially for comparing LLM-based approach with traditional approaches like CNN.

**Ethical Concerns:**

["NO or VERY MINOR ethics concerns only"]

**Final Justification:**

The authors have addressed my concerns, and I'll maintain my recommendation towards accepting the paper.

**Limitations:**

Yes

**Quality:**

3

**Strengths And Weaknesses:**

This paper is commended for its clarity, strong motivation, and technical accuracy. A notable strength is its thorough empirical analysis, which involves a systematic comparison across diverse prompts and large language models. The consistency of these results significantly bolsters the paper's conclusions.

A weakness of the paper is that its proposed solution, particularly for the localization issue, appears rather simple and established. It is known for the LLM community that LLMs perform poorly at localizing the location of  a sequence of numbers (,e.g., performing carrying operations in multi-digit addition). The proposed solution, index-aware prompting, also lacks novelty. However, the experimental results still shows that this simple approach provides significant gains, eliminating the need of novel (and perhaps more complicated approach). The work can still offer important insights to the community.

---

> ### Author Rebuttal · Authors · 2025-07-30
>
> We sincerely appreciate the reviewer K63Q’s thoughtful and constructive feedback. The comments have been invaluable, and we would like to carefully address the remaining concerns, particularly the simplicity of our approach and the need for deeper analysis of its limitations.
>
>
> > **W1) Proposed solution, particularly for the localization issue, appears rather simple and established.**
>
> We agree that incorporating explicit positional indices is conceptually straightforward and builds on established ideas, such as those presented by Liu et al. [1].
>
> That said, as the reviewer K63Q also pointed out, even this simple approach leads to significant and consistent performance improvements in the context of LLM-based TSAD. Notably, this strategy has been largely overlooked in prior work, often without a clear analysis of why LLMs struggle in this setting. Our contribution lies not in introducing a novel technique, but in identifying localization as a fundamental bottleneck and showing that a lightweight intervention can effectively address it. As demonstrated in Table 3, this results in substantial performance gains across models.
>
> We believe this finding highlights an important insight: even simple modifications, when carefully aligned with underlying failure modes, can lead to unexpectedly large performance improvements. We hope this encourages further investigation into the limitations of LLMs and inspires the development of both simple and more principled solutions.
>
> [1] Large language models can deliver accurate and interpretable time series anomaly detection, arXiv’2405
>
> > **Q1) It would be good to highlight some failure cases of the proposed approach.**
>
> We appreciate the reviewer K63Q’s suggestion to highlight failure cases of our proposed method, especially in comparison with conventional approaches such as the CNN-based detection model.
> In line with this, we have carefully analyzed our results on the TSB-AD-U benchmark and shared several failure cases that reveal the limitations of LLM-based TSAD under specific conditions:
>
> * _Decomposition challenges with non-linear patterns:_
> Our method relies on statistical decomposition to remove seasonality, which inherently assumes linearity. However, the MSL time series in the TSB-AD-U benchmark includes anomaly patterns that exhibit non-linear and irregular structures. As a result, the decomposition step may fail to fully isolate seasonal components, leading to residual distortions and false positives that ultimately degrade performance. In contrast, semi-supervised models such as the CNN-AD and the AutoEncoder, which are trained end-to-end, often learn to accommodate such complexities implicitly. Consequently, while our method outperforms the CNN overall on the TSB-AD-U benchmark, the CNN achieves slightly better F1 score on the MSL time series specifically.
>
> * _Limitations with long sequence visualization:_
> For long sequences with densely packed patterns (e.g., TAO time series in TSB-AD-U), converting the input into a single image plot compresses local patterns, making them visually indistinct. This hinders the LLM’s ability to recognize anomalies from the plot. The CNN model that processes raw sequences or local windows can be more robust in such scenarios, as they are not constrained by the spatial resolution of a visual representation. While LLMs could, in principle, also process the sequence in smaller local windows and aggregate the predictions, this significantly increases computational cost—already a known drawback of LLM-based approaches—and may not be practical for real-time or resource-constrained scenarios.
>
> We believe these failure cases are important for understanding both the strengths and limitations of LLMs. To reflect this, we include representative qualitative examples in the revised version. While LLMs provide clear benefits such as semantic reasoning and the ability to generalize to new tasks without additional training, they are not a universal solution. As the reviewer suggests, recognizing both their potential and their limitations is essential for guiding future research in this area.

---

> > ### Comment · Reviewer_K63Q · 2025-08-04
> >
> > Thank you to the authors for the thoughtful response! I am maintaining my score in favor of acceptance.

---

> > > ### Author Response · Authors · 2025-08-08
> > >
> > > We sincerely appreciate your constructive feedback and positive evaluation. We will make sure to thoughtfully incorporate your suggestions into the revised version.

---

### Official Review · Reviewer_NVeG · 2025-07-01

**Clarity:** 2
**Significance:** 3
**Originality:** 3
**Rating:** 4
**Confidence:** 3

**Summary:**

This paper identifies two challenges, temporal understanding and anomaly localization, in LLM-based time series anomaly detection. Correspondingly, the authors proposes a novel method to solve these challenges. Comprehensive experimental results support the claim of this paper.

**Questions:**

Q1. In Fig 2, the types of different anomalies should be further explained.
Q2. While the motivation behind Section 2.2 is well, the experiment results could be further explained. Is this setting identical to directly query LLMs to return the coordinates given one image with highlighted anomaly region? Intuitively, LLMs should perfectly give the results. But why there is still a big margin to the perfect prediction? The authors need to give a convincing reason.
Q3. The claim in line 143 to 146 should be theoretically supported or experimentally supported. Or recent related findings by others should be provided.
Q4. The authors need to give more details of the three steps of Section 3.1. What is the role of ACF function? How can step 2 be implemented?
Q5. The experiment settings of Section 4.2.2 should be further explained. How does each variant be implemented?
Q6. The provided experiment in Section 5.2 should be further explained. What is the difference between temporal context and pattern context here? How does Fig 8 support the claim in Section 5.2?

**Ethical Concerns:**

["NO or VERY MINOR ethics concerns only"]

**Final Justification:**

I recommend Borderline accept. My main concerns have been addressed after author rebuttal.

**Limitations:**

yes

**Quality:**

3

**Strengths And Weaknesses:**

Strengths:
S1. The research topic is important, and the proposed method is novel.
S2. The pre-experiment w.r.t. the analysis for existing challenges is well-movitated.
S3. The paper is well-written and easy to follow.

Weakness:
W1. In Fig 2, the types of different anomalies should be further explained.
W2. While the motivation behind Section 2.2 is well, the experiment results could be further explained.
W3. The claim in line 143 to 146 should be theoretically supported or experimentally supported.
W4. The authors need to give more details of the three steps of Section 3.1. What is the role of ACF function? How can step 2 be implemented?

The details and corresponding questions about these weaknesses can be found in Questions below.

---

> ### Author Rebuttal · Authors · 2025-07-31
>
> We appreciate the reviewer NVeG’s constructive feedback and have made our best effort to address the concerns thoroughly.
>
> > **W1, Q1) In Fig 2, the types of different anomalies should be further explained.**
>
> While the definitions of each anomaly type are provided in Appendix B.2, we acknowledge that including a concise explanation in Section 2.1, would improve clarity and self-containment. To address this, we briefly describe the four anomaly types as follows:
>
> * _Contextual Point_: a single timestamp shows a sharp deviation from its neighbors.
> * _Out of Range_: a contiguous segment of values deviates significantly from the expected pattern.
> * _Trend Shift_: a structural shift in the overall slope or direction of the time series.
> * _Frequency Change_: a disruption in the expected periodicity, such as altered oscillation rate.
>
> These anomaly types, based on the AnomLLM benchmark, reflect diverse real-world patterns. Point and range anomalies show clear, sudden deviations. In contrast, trend and frequency anomalies involve subtle changes like slow drifts or irregular cycles within normal ranges. These are harder to catch with simple rules or local statistics and need models that understand temporal dynamics. This variety supports a balanced and thorough evaluation.
>
>
> > **W2,Q2) Clarification for the large gap from perfect prediction in Figure 3**.
>
> Your understanding of the experimental setting in Section 2.2 is correct. In this setup, the LLM is provided with a time-series image where the anomaly region is visually highlighted, and the task is to return the corresponding start and end coordinates. This isolates the localization task by removing the need for anomaly understanding.
>
> However, even with the anomaly region clearly marked, localization remains difficult. The main reason is the sparse index visibility in plots. For long time series (e.g., 1000 timesteps), it’s not feasible to label every timestamp along the x-axis without clutter. This means the LLM cannot directly "read" exact indices from the image. Instead, it must infer coordinates based on the relative position of the highlighted region using only a few reference ticks. This indirect reasoning is error-prone, especially for small or subtle anomalies.
>
> This challenge explains the performance gap observed in Section 2.2 and directly motivates one of the core design choices in our method, which is index-aware prompting. By explicitly attaching each value with its index in the time-series text, we remove the need for visual position inference and turn localization into a retrieval task.
>
>
> > **W3 Q3) The claim in line 143 to 146 should be supported.**
>
> In response to the reviewer’s thoughtful suggestion, we conducted additional experiments using LoRA[1] fine-tuning on the Qwen2.5 and LLaMA3 models to provide empirical evidence for lines 143–146. The results show that while fine-tuning enhances the models' decomposition capabilities, it significantly degrades their performance on the MMLU benchmark, which evaluates general reasoning ability.
>
> **Rebuttal-Table 3: Impact of LoRA fine-tuning on component detection task, highlighting the trade-off between task adaptation and reasoning retention.**
> |                        | Task        | Component |  Detection      |    | |    Reasoning      |        |
> |-|-|-|-|-|-|-|-|
> | Model                  | Fine-tuning | Precision                | Recall | F1   |           | MMLU_val | MMMU   |
> | LLaMA3.1-8B-Instruct   | X           |                     0.61 |   0.58 | 0.59 |           | 68.20    | -      |
> | LLaMA3.1-8B-Instruct   | LoRA        |                     0.67 |   0.67 | 0.67 |           | 58.45    | -      |
> |    **Δ (LoRA - Base)**        | -           |                     **0.06** |   **0.09** | **0.08** |           | **-9.74**    | -      |
> | Qwen2.5-VL-3B-Instruct | X           |                     0.35 |   0.59 | 0.44 |           | 65.20    | 43.11  |
> | Qwen2.5-VL-3B-Instruct | LoRA        |                     0.76 |   0.76 | 0.75 |           | 63.79    | 32.00  |
> |   **Δ (LoRA - Base)**        |      -       |                     **0.41** |   **0.17** | **0.31** |           | **-1.41**    | **-11.11** |
>
> These findings are in line with prior studies [2, 3, 4, 5], which have reported the difficulty of extending LLMs to new domains without compromising their original capabilities. Our results thus provide empirical support for adopting external decomposition tools, allowing LLMs to focus on their reasoning over structured input. We will include these findings in the revised version of the paper.
>
> [1] LoRA: Low-Rank Adaptation of Large Language Models, ICLR'22
>
> [2] An Empirical Study of Catastrophic Forgetting in Large Language Models During Continual Fine-tuning, arXiv’2308
>
> [3] Mitigating Catastrophic Forgetting in Large Language Models with Self-Synthesized Rehearsal, ACL’24
>
> [4] Revisiting Catastrophic Forgetting in Large Language Model Tuning, ACL’24
>
> [5] Language Model Alignment with Elastic Reset, NeurIPS’23
>
>
> > **W4, Q4) More details of the three steps of Section 3.1: the role of ACF function and the implementation of step 2.**
>
> #### (1) Implementation Details of Step 2
> In practice, this decomposition is implemented using the _statsmodels.seasonal_decompose_ function with _model='additive'_ and the estimated period $ P^\ast $ as the seasonal window. The full implementation is provided in the supplementary code.
>
> Formally, given a time series $  \\{x_t\\}_{t=1}^T  $ and the dominant seasonal period $  P^\ast $ identified via the ACF (see below), we decompose the series as: $x_t = \tau_t + s_t + r_t, $ where $ \tau_t $ is the trend, $ s_t $ is the seasonal component, and $ r_t $ is the residual.
>
> The trend is estimated using a center moving average over window size $ P^\ast $:
>
> $$
> \tau_t = \frac{1}{P^\ast} \sum_{i = -\lfloor P^\ast/2 \rfloor}^{\lfloor P^\ast/2 \rfloor} x_{t+i}.
> $$
>
> For the seasonality, we compute an average of the de-trended values aligned by their phase in the seasonal cycle:
>
> $$
> s_t = \frac{1}{N_c} \sum_{j \in \mathcal{I}_c} (x_j - \tau_j), \quad \text{where } c = t \bmod P^\ast,\; \mathcal{I}_c = \{ j \mid j \bmod P^\ast = c \}.
> $$
>
> This results in a seasonal template $ \\{s_1, \dots, s_{P^\ast}\\} $, which we repeat to obtain $ \\{s_t\\}_{t=1}^T $.
> The residual is then computed as: $r_t = x_t - \tau_t - s_t.$
>
> #### (2) Role of ACF in the Decomposition Pipeline
> The ACF plays a critical role in estimating the seasonal period $ P^\ast $, which directly determines the window size for trend estimation and the cycle length for seasonality computation.
>
> We compute the sample ACF as:
>
> $$
> \hat{\rho}(k) = \frac{1}{T - k} \sum_{t=1}^{T-k} \frac{(x_t - \bar{x})(x_{t+k} - \bar{x})}{\hat{\sigma}^2},
> $$
> and select the dominant period as:
> $$
> P^\ast = \mathop{\mathrm{argmax}}_{k \in [1, T/2]}{ \hat{\rho}(k)}.
> $$
>
> By aligning decomposition with the dominant autocorrelated lag, the periodic structure is removed from the input. This prevents seasonal leakage into the residual component and allows LLMs to focus on irregular or subtle anomalies. We will clarify these implementation details and rationale in the revised version of Section 3.1.
>
>
> > **Q5) The details of experiment setting in Section 4.2.2.**
>
> We apologize for not properly guiding readers to the detailed implementation settings, which are provided in the appendix rather than in the main text.
>
> Section 4.2.2 evaluates the time-series decomposition capabilities of LLMs through two tasks: component detection and component generation. Supplementary descriptions of these tasks are provided in Appendix D.4, and the full prompt templates are shown in Figure 17.
>
> In the component detection task, we formulate the problem as a multi-label classification setup using QA-style prompts, where the model is asked whether trend and/or seasonality are present. As ground truth, we use synthetic time series constructed with known components. To evaluate the LLM’s predictions, we apply a simple statistical decomposition method and label a component as “present” if its min-max amplitude range exceeds a fixed percentage of the original series range. This provides a consistent rule-based reference against which the LLM’s classification output can be compared.
>
> In the component generation task, LLMs are prompted to produce the trend or seasonal component as a numeric sequence. This setup is structurally similar to LLM-based forecasting prompts, where the model generates a sequence conditioned on input. For evaluation, we compare the sequence generated by the LLM or the statistical decomposition to the ground-truth component defined during synthetic data generation, using Mean Absolute Error.
>
> > **Q6) The details of experiment in Section 5.2.**
>
> In Section 5.2, we highlight a key strength of LLMs: their ability to perform context-aware anomaly filtering. We distinguish between two types of domain-specific context: _temporal context_, which refers to time-based events such as scheduled maintenance periods, and _pattern context_, which describes known behaviors in the time-series shape, such as occasional flat segments caused by sensor resets. By leveraging these contextual cues expressed in natural language, LLMs can selectively exclude explainable anomalies and focus on unexpected ones that align with the semantic definition of novelty.
>
> Figure 8 supports this claim by qualitatively demonstrating how LLM predictions differ with and without contextual information. LLMs can suppress such explainable patterns when justified by the given context (e.g., power restrictions or system shutdowns).
>
> However, this result remains qualitative and relies on simulated annotations. Therefore, there is a clear need for context-annotated TSAD benchmarks to enable rigorous quantitative evaluation. We see this as a promising avenue for future work and aim to further explore the unique reasoning capabilities of LLMs in time-series analysis.

---

> > ### Comment · Reviewer_NVeG · 2025-08-03
> >
> > Thanks for the authors for the detailed response. My main concerns have been addressed.

---

> > > ### Author Response · Authors · 2025-08-08
> > >
> > > We sincerely appreciate your review of our rebuttal. As we mentioned earlier, we will thoroughly explore and incorporate the valuable suggestions you have kindly provided.

---

### Official Review · Reviewer_kXX6 · 2025-07-03

**Clarity:** 4
**Significance:** 3
**Originality:** 4
**Rating:** 5
**Confidence:** 4

**Summary:**

This study examines LLM usage in the context of time-series anomaly detection (TSAD). The authors explored why LLMs are prone to fail TSAD tasks and proposed improving prompting strategies based on the removal of seasonality and index-aware prompting. In addition, the authors also compared LLM-based TSAD strategies against traditional non-LLM-based approaches. They found that an LLM approach achieved the overall highest F1 score and can take advantage of context to isolate anomalies.

**Questions:**

The analysis in the introduction provided a detailed explanation of "how" LLMs are failing the TSAD tasks, but not "why". It is challenging to offer interpretability for the opaque black box that is an LLM, but this manuscript could benefit greatly from an additional interpretability analysis, such as delving into the LLM's attention head importance, etc.

Using the proposed prompting strategy improved the performance significantly on some models (e.g., GPT-4o) but only slightly on others (e.g., LLaMA 3). What are the implications of these inconsistencies across different models? Is there a common characteristic among the high-performing models? What LLM should a person ultimately choose when facing a TSAD task?

**Ethical Concerns:**

["NO or VERY MINOR ethics concerns only"]

**Final Justification:**

I appreciate the authors taking the time and effort to interpret LLM behavior and perform the additional evaluations on LLM types. My original concerns are addressed.

**Limitations:**

yes

**Paper Formatting Concerns:**

No particular formatting concerns

**Quality:**

3

**Strengths And Weaknesses:**

Quality: The study is, in general, well supported by experimental results and evidence. However, I am not convinced that it answered the question "WHY LLMs fail at TSAD." Instead, the manuscript explains clearly "HOW LLMs fail at TSAD." by observing the pattern that LLMs can fail at certain types of anomalies based on prompt types and can fail at identifying anomaly locations. As to why these observations happen, the manuscript does not offer a satisfying explanation. I recommend either changing the wording from "why" to "how" or digging deeper to find out the root cause of these observations.
Clarity: The manuscript is clearly written and well organized.
Significance: This study provides a better data preprocessing and prompting strategy to drastically improve the performance of LLM on the task of TSAD. It provides a use case of zero-shot TSAD using an LLM that has a wide range of real-world applications. However, I am concerned by the results that this prompting strategy has a varying degree of improvement across different LLMs. The manuscript does not provide an explanation for why using this on GPT-4o resulted in much higher improvement than using it on LLaMA 3.
Originality: This study provides a new time-series prompting strategy that outperforms established methods. It also provides a comparison with non-LLM methods in TSAD, which is lacking in current research.

---

> ### Author Rebuttal · Authors · 2025-07-31
>
> > **Q1) The analysis in the introduction provided a detailed explanation of "how" LLMs are failing the TSAD tasks, but not "why". It is challenging to offer interpretability for the opaque black box that is an LLM, but this manuscript could benefit greatly from an additional interpretability analysis, such as delving into the LLM's attention head importance, etc.**
>
> We thank the reviewer kXX6 for the insightful feedback. We fully agree that a more in‑depth analysis of interpretability can better illuminate the underlying mechanisms in our study.
> Due to the limited rebuttal period and resource constraints, we were unable to conduct experiments at the same scale as those presented in the main paper. However, we share an important finding based on an analysis of attention weights to address the reviewer's concern.
>
> The deseasonalization technique we proposed aims to mitigate the issue of subtle anomalies being obscured by dominant seasonal patterns. We hypothesize that if deseasonalization effectively removes such dominant seasonal components, LLMs will be better able to attend to subtle anomalies. Consequently, we expect the attention weights over anomalous regions to vary depending on whether deseasonalization is applied. Specifically, we use the phrase "Anomalous pattern" as the query and the textual time series (i.e., the sequence in the prompt) as the key. The table below presents the results of this experiment on trend and frequency anomaly datasets. As a metric, _Increased Sample Ratio_ refers to the proportion of samples in which the attention weight on the anomaly segments increased after removing seasonality. _Avg. Attention Improvement_ indicates the overall average of the increased attention weights ratio per sample compared to before.
>
> **Rebuttal-Table 1. Attention analysis results showing that deseasonalization increases LLM focus on anomalous regions.**
> | Model                  | Increased Sample Ratio (Trend) | Avg. Attention Improvement (Trend) | Increased Sample Ratio (Freq.) | Avg. Attention Improvement (Freq.) |
> |------------------------|-------------------------------|------------------------------------|-------------------------------|------------------------------------|
> | LLaMA-3.1-8B-Instruct  | 82.35%                        | 28.91%                             | 77.51%                        | 120.85%                            |
> | Qwen2.5-3B-Instruct    | 77.64%                        | 23.51%                             | 71.98%                        | 99.75%                             |
>
> As shown in the table, deseasonalization consistently enhances attention to anomalous regions across both Trend and Freq. anomaly datasets.
>
> For example, LLaMA-3.1-8B-Instruct showed increased attention in 82.35% of Trend samples (28.91% avg. gain) and 77.51% of Freq. samples (120.85% avg. gain). Similarly, Qwen2.5-3B-Instruct exhibited increases in 77.64% of Trend samples (23.51% avg. gain) and 71.98% of Freq. samples (99.75% avg. gain). These results indicate that deseasonalization helps LLMs better focus on subtle anomalies by reducing dominant seasonal noise, boosting detection performance.
>
> Apart from the in-depth experiments, we would like to carefully explain our view on the distinction between “why” and “how”. Prior works applying LLMs to TSAD have primarily focused on performance evaluation, often using observational analyses to characterize **how these models fail** (e.g., LLMs fail more frequently on trend and frequency anomalies than on point and range anomalies), with few explicitly exploring the underlying mechanisms. In contrast, our paper takes a step toward answering **why LLMs underperform in TSAD** by explicitly identifying two causes: (1) insufficient time‑series understanding and (2) limited localization (counting) ability. We uncover these two core limitations and propose prompt‑design strategies that directly target them. These strategies lead to measurable performance improvements, which we believe constitute a practical and meaningful contribution beyond prior work.
>
> However, we still appreciate the reviewer’s point that what we call “why” can, at a deeper level, still be interpreted as another form of “how”. We view this as intrinsic to scientific inquiry: as one probes deeper causal mechanisms, new questions naturally emerge. For example,  following the reviewer’s suggestion, we performed an interpretation of attention heads and qualitatively confirmed that, in the absence of our proposed deseasonalization, attention tends to be absorbed by seasonal patterns. This finding naturally leads to a deeper "why", prompting questions about the origin of specific attention patterns. In turn, previous explanations become another form of "how".
>
> Ultimately, our contribution provides insights into underlying causes of failure that were not previously addressed in the literature, laying groundwork for deeper analyses suggested by the reviewer. We agree that further interpretability studies, such as more extensive analyses of attention heads and embedding dynamics, are highly valuable directions as future research topics. Once again, we are grateful for the reviewer’s perspective; this feedback will serve as important guidance for our subsequent research.
>
>
> > **Q2) Using the proposed prompting strategy improved the performance significantly on some models (e.g., GPT-4o) but only slightly on others (e.g., LLaMA3). What are the implications of these inconsistencies across different models? Is there a common characteristic among the high-performing models? What LLM should a person ultimately choose when facing a TSAD task?**
>
> As the reviewer pointed out, despite applying the same prompting strategy, the degree of performance improvement varied across models. Our proposed method led to meaningful performance gains across all evaluated models, including GPT-4o, Qwen, Gemini, and LLaMA3. Nevertheless, even under identical settings, GPT-4o, Qwen, and Gemini exhibited substantially greater improvement than LLaMA3.
>
> We fully agree that this discrepancy warrants further analysis. To better understand this gap, we investigated the common characteristics of the higher-performing models by analyzing the correlation between their general reasoning ability and TSAD performance. Specifically, we compared the models using MMLU (for language reasoning) and MMMU (for visual reasoning), which are representative benchmarks for multimodal LLMs.
>
> **Rebuttal-Table 2. Comparison of TSAD performance gains across LLMs, highlighting the relationship between general reasoning ability and TSAD improvement.**
> | Benchmark        | Method    | GPT-4o | Qwen2.5-VL-72B-Instruct | Gemini-1.5-Flash-002 | InternVL2-LLaMA3-76B |
> |------------------|-----------|--------|---------------------------|-----------------------|----------------------|
> | TSAD(F1)          | AnomLLM   | 42.03  | 46.65                    | 43.56                | 33.67               |
> | TSAD(F1)          | Our       | 70.04  | 67.56                    | 63.31                | 34.66               |
> | MMMU             | -         | 69.10  | 64.50                    | 56.10                | 58.20               |
> | MMLU-Pro         | -         | 74.68  | 71.59                    | 64.09                | 56.20               |
> | MMLU-Pro (STEM + Economics) |           | 72.83  | 73.36                    | 63.60                | 52.50               |
> | MMLU-Pro (Others)           |           | 76.50  | 70.61                    | 64.98                | 61.27               |
> * We used the scores from the Hugging Face MMLU-Pro leaderboard and the official MMMU leaderboard. For MMLU-Pro, we referred to the corresponding language-only models of each architecture (e.g., Qwen2.5-72B-Instruct, LLaMA3-70B-Instruct).
>
> Interestingly, within the limited scope of our study, models achieving higher scores on MMLU and MMMU benchmarks also showed greater TSAD improvements. For example, GPT-4o (MMLU 74.68, MMMU 69.10), Qwen (71.59, 64.50), and Gemini (64.09, 56.10) all saw large F1 improvements, +28.01, +20.91, and +19.75 points respectively. In contrast, Intern-VLM, with lower scores on both MMLU (56.20) and MMMU (58.20), showed only a marginal TSAD improvement of +0.99.
>
> We then looked more closely at numerically demanding subdomains of MMLU, such as STEM (math, physics, chemistry, engineering) and Economics. These subdomains are especially relevant to TSAD, which often requires numerical pattern recognition and quantitative reasoning. In these areas, the performance gap between LLaMA3 and the other models became even more pronounced: GPT-4o (72.83), Qwen (73.36), and Gemini (63.60) significantly outperformed LLaMA3 (52.50), widening the relative gap seen in overall MMLU.
>
> Lastly, we also considered the reviewer’s practical question: “Which LLM should one choose for TSAD tasks?” Since running full TSAD benchmarks for each new LLM is impractical, we suggest using numerically-oriented MMLU subcategories as a reasonable proxy for TSAD capability. Our empirical results support this idea, indicating that MMLU scores can partially predict a model’s effectiveness in TSAD, offering a practical and scalable alternative to direct TSAD benchmarking.

---

### Note · Authors · 2025-08-14

### **Dear Reviewers and Area Chairs**

We sincerely thank Reviewers kXX6, NVeG, K63Q, and 7DTS for their constructive and insightful feedback. Our work analyzes the reasons for performance degradation of LLMs in Time-Series Anomaly Detection and identifies two key limitations: insufficient temporal understanding and limited localization ability. To address these, we propose **deseasonalization** and **index-aware prompting**, which consistently improve performance across various models and datasets.

During the rebuttal period, we responded to the reviewers’ concerns and questions as follows, and we will incorporate these points into the revised paper:

- Provided **attention analysis** demonstrating that deseasonalization improves focus on anomalous regions.
- Analyzed **cross-model performance variation** and showed that numerically oriented MMLU subdomains can serve as useful indicators for predicting TSAD performance.
- Clarified the **decomposition procedure**, the role of the ACF function, and prompt implementation details.
- Included **failure cases** where conventional methods such as CNNs outperform LLM-based approaches, and explained the differences between detection and localization performance.

These points will all be reflected in the revised version, which will further enhance the completeness and practical value of the work. We appreciate the reviewers’ acknowledgment that the main concerns have been addressed, and we believe these revisions will further enhance the paper’s value and usefulness for both researchers and practitioners.

---

### Decision · Program_Chairs · 2025-09-17

**Decision:**

Accept (poster)

**Comment:**

After the final round of reviews, the paper received one borderline accept recommendation and three accept recommendations. The reviewers highlighted the relevance and timeliness of the topic, particularly the application of LLM to time series anomaly detection. They also recognized the paper's technical contributions as a strength. The rebuttal was viewed positively and contributed to an improved overall assessment, as it addressed several of the reviewers’ concerns and provided clarification on key points related to the methodology and experimental results.